# Forget by Uncertainty: Orthogonal Entropy Unlearning for Quantized Neural Networks

**Tian Zhang** [* 1] **Yujia Tong** [* 1] **Junhao Dong** [2] **Ke Xu** [1] **Yuze Wang** [1] **Jingling Yuan** [1]

## Abstract

The deployment of quantized neural networks on edge devices, combined with privacy regulations like GDPR, creates an urgent need for machine unlearning in quantized models. However, existing methods face critical challenges: they induce forgetting by training models to memorize incorrect labels, conflating forgetting with misremembering, and employ scalar gradient reweighting that cannot resolve directional conflicts between gradients. We propose **OEU**, a novel Orthogonal Entropy Unlearning framework with two key innovations: 1) Entropy-guided unlearning provides an unbiased forgetting direction by maximizing prediction uncertainty on forgotten data, avoiding confident misprediction toward any specific class, and 2) Gradient orthogonal projection eliminates interference by projecting forgetting gradients onto the orthogonal complement of retain gradients, providing theoretical guarantees for utility preservation under first-order approximation. Extensive experiments demonstrate that OEU outperforms existing methods in both forgetting effectiveness and retain accuracy.

## 1. Introduction

Model quantization is indispensable for deploying deep neural networks on edge devices (Howard et al., 2017; Zhou et al., 2018), reducing computational overhead through low-bit parameter representations. While this technique enables efficient on-device inference, data protection legislation, including the General Data Protection Regulation (GDPR) (Hoofnagle et al., 2019), has introduced new challenges for quantized neural networks (QNNs). These regulations establish the *"right to be forgotten"*, granting users the legal authority to demand the erasure of their data from machine learning systems (Nguyen et al., 2025). Consequently, implementing effective machine unlearning in QNNs has become an urgent yet underexplored challenge.

Machine unlearning (MU) offers a solution by removing specific data influences from trained models without full retraining (Nguyen et al., 2025; Dong et al., 2025b; Mavrothalassitis et al., 2025). Existing approaches include exact unlearning (Bourtoule et al., 2021), which retrains on retained data but incurs high computational cost, and approximate unlearning (He et al., 2025), which efficiently modifies model parameters to approximate the retrained state. Approximate methods, *e.g.*, gradient ascent (Graves et al., 2021; Thudi et al., 2022; Dong et al., 2025a), influence functions (Koh & Liang, 2017; Izzo et al., 2021), and random label (Golatkar et al., 2020) assignment show promising results, yet they are designed for full-precision models and assume continuous parameter spaces. These assumptions fail in quantized networks, where discrete weight representations introduce unique challenges. Q-MUL (Tong et al., 2025) emerged as the first unlearning framework for QNNs, employing Similar Labels to reduce noise injection and Adaptive Gradient Reweighting to balance forgetting and retain gradients.

Despite its pioneering contributions, Q-MUL (Tong et al., 2025) exhibits two fundamental limitations that hinder its unlearning effectiveness. *First, the Similar Labels strategy still adheres to the paradigm of "learning incorrect answers."* As illustrated in Figure 1(a), the original model correctly assigns high confidence to the true label (*class* 0). However, the Similar Labels approach simply shifts this confidence to an alternative wrong label (*class* 1), essentially training the model to memorize this mislabel. This conflates forgetting" with "remembering incorrectly," potentially introducing systematic biases toward specific classes and ultimately failing to achieve genuine data influence removal.

*Second, Adaptive Gradient Reweighting merely adjusts the scalar magnitudes of gradients while entirely neglecting their directional conflicts.* As depicted in Figure 1(b), when the angle between the forgetting gradient and retain gradient exceeds $90°$, any update along the forgetting direction

---

*Equal contribution [1] Hubei Key Laboratory of Transportation Internet of Things, School of Computer Science and Artificial Intelligence, Wuhan University of Technology, China. [2] College of Computing and Data Science, Nanyang Technological University, Singapore. Correspondence to: Yujia Tong <tyjjjj@whut.edu.cn>.

*Proceedings of the 43$^{rd}$ International Conference on Machine Learning*, Seoul, South Korea. PMLR 306, 2026. Copyright 2026 by the author(s).

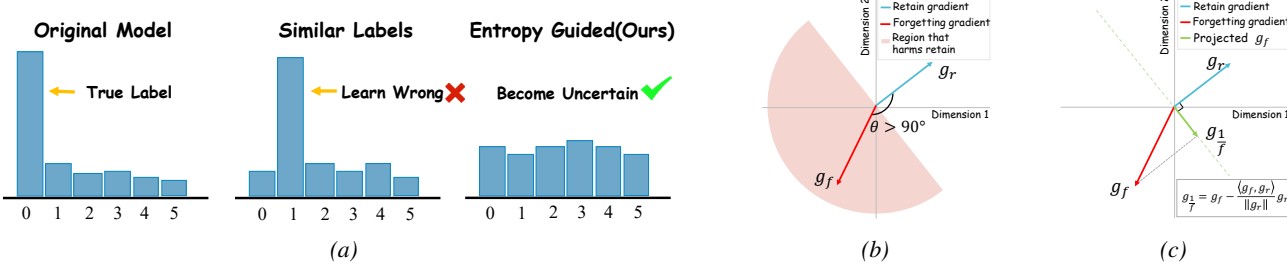

*Figure 1.* Motivation of OEU. (a) Similar Labels shifts confidence to a wrong class, while our entropy-guided approach drives predictions away from confident outputs without class-specific bias. (b) Directional conflict between forgetting and retain gradients. (c) Orthogonal projection eliminates this conflict.

inevitably harms retain performance—no scalar weighting scheme can ever prevent this. This directional conflict is particularly pronounced in QNNs, where the constrained parameter space amplifies gradient interference.

These limitations motivate us to fundamentally rethink the challenges of unlearning in QNNs:

> **Challenges:**
>
> - **C1: What should be the unlearning objective?** Existing methods train models to memorize incorrect labels, conflating "forgetting" with "misremembering" and introducing systematic biases.
>
> - **C2: How to optimize without harming retain performance?** Scalar gradient reweighting cannot resolve directional conflicts between forgetting and retain gradients, especially in constrained quantized parameter spaces.

To address these challenges, we propose OEU, a novel Orthogonal Entropy Unlearning framework with two innovations. For **C1**, we reconceptualize the unlearning objective through an entropy-guided perspective: instead of training the model to produce incorrect predictions, we maximize output entropy to provide an unbiased forgetting direction, guiding predictions away from confident outputs without introducing class-specific biases, as shown in Figure 1(a). For **C2**, we propose gradient orthogonal projection to resolve directional conflicts. As illustrated in Figure 1(c), by projecting the forgetting gradient onto the orthogonal complement of the retain gradient, we mathematically guarantee that unlearning updates do not interfere with retained knowledge. Together, maximum entropy defines what to forget, and orthogonal projection ensures how to forget without collateral damage. We summarize our contributions below:

- We challenge the prevailing "learn incorrect labels" approach and propose entropy-guided unlearning (EGU), , which provides an unbiased forgetting direction by maximizing prediction uncertainty on forgotten data.

- We introduce gradient orthogonal projection (GOP) to eliminate directional interference between forgetting and retain gradients, providing theoretical guarantees for utility preservation under first-order approximation.

- Extensive evaluations across multiple datasets and quantization configurations demonstrate that OEU consistently outperforms existing methods in both forgetting effectiveness and retain accuracy.

## 2. Related Work

**Model Quantization** has emerged as a fundamental technique for deploying deep neural networks on resource-constrained devices by reducing the bit-width of weights and activations. Existing methods include Post-Training Quantization (PTQ) (Nagel et al., 2020; Cai et al., 2020; Zhang et al., 2024), which directly quantizes pre-trained models but often suffers accuracy degradation at low bit-widths, and Quantization-Aware Training (QAT), which simulates quantization during training to better adapt to quantization-induced errors. QAT relies on the Straight-Through Estimator (STE) (Bengio et al., 2013) for gradient backpropagation through non-differentiable operations, with subsequent advances including learnable clipping thresholds (Choi et al., 2018), differentiable soft quantization (Gong et al., 2019), and learnable step sizes (Esser et al., 2019; Bhalgat et al., 2020; Li et al., 2024). These techniques have enabled widespread deployment of QNNs on edge devices (Bruschi et al., 2020; Shen et al., 2024; Tong et al., 2026). However, as user data becomes increasingly entangled with model parameters, significant privacy concerns arise—a challenge that existing quantization literature has largely overlooked. Our work addresses this gap by investigating machine unlearning specifically designed for QNNs.

**Machine Unlearning** aims to remove the influence of specific training data from learned models, motivated by privacy regulations such as GDPR (Hoofnagle et al., 2019) that grant individuals the "right to be forgotten." The gold standard is retraining from scratch on retained data, but this remains

computationally prohibitive for large-scale models. Approximate methods have been developed to efficiently modify model parameters, including influence function-based approaches (Koh & Liang, 2017; Izzo et al., 2021), gradient ascent methods (Graves et al., 2021; Thudi et al., 2022), and label manipulation techniques such as Random Labels (Golatkar et al., 2020), $\ell_1$-sparse (Jia et al., 2023), and SalUn (Fan et al., 2024). However, these methods are designed for full-precision models and operate under a "learning wrong answers" paradigm that may introduce systematic biases. Q-MUL (Tong et al., 2025) is the first work addressing unlearning for QNNs, but it inherits the same paradigm and only adjusts gradient magnitudes without resolving directional conflicts. Our work departs from this paradigm by proposing EGU for genuine uncertainty, complemented by GOP to eliminate directional conflicts.

## 3. Preliminaries

In this section, we establish the foundational concepts for our work by revisiting machine unlearning and Quantization-Aware Training (QAT). As MU methods typically require gradient-based optimization, unlearning in QNNs can be naturally formulated as a QAT process.

### 3.1. Revisiting Machine Unlearning

Let $D = \{(x_i, y_i)\}_{i=1}^N$ denote the complete training dataset with $N$ samples, where $x_i \in \mathbb{R}^d$ represents the $i$-th input sample and $y_i \in \{1, 2, \ldots, K\}$ is the corresponding class label. The forgotten dataset $D_f \subseteq D$ represents the subset of data that needs to be removed from the trained model, while the retained dataset $D_r$ denotes its complement, satisfying $D_f \cap D_r = \varnothing$ and $D_f \cup D_r = D$. Based on the composition of $D_f$, machine unlearning in image classification can be categorized into two scenarios: class-wise forgetting, where $D_f$ consists entirely of samples from specific classes to eliminate, and random data forgetting, where $D_f$ contains randomly selected samples from one or multiple classes.

We denote the original model trained on $D$ as $\mathcal{M}_0$ with parameters $\theta_0$. The retrained model $\mathcal{M}_r$, obtained by training from scratch solely on the retained dataset $D_r$, serves as the "gold standard" for unlearning evaluation (Nguyen et al., 2025). However, retraining incurs substantial computational overhead, particularly for large-scale models. Approximate unlearning methods aim to obtain an unlearned model $\mathcal{M}_u$ by efficiently modifying the original model $\mathcal{M}_0$ to eliminate the influence of $D_f$, such that $\mathcal{M}_u$ approximates $\mathcal{M}_r$ in terms of both prediction behavior and privacy guarantees. The effectiveness of unlearning is typically evaluated through multiple metrics: Forget Accuracy (FA) measuring performance on $D_f$, Retain Accuracy (RA) on $D_r$, Test Accuracy (TA) on held-out test data, and Membership Inference Attack (MIA) success rate assessing privacy

leakage (Jia et al., 2023; Fan et al., 2024; Tong et al., 2025).

### 3.2. Revisiting Quantization-Aware Training

Quantization-Aware Training (QAT) simulates quantization effects during training by inserting fake quantization nodes that perform quantization and dequantization operations on floating-point values. For $n$-bit signed quantization with scale factor $s$, the quantization function $q(\cdot)$ is defined as:

$$x^q = q(x^r) = s \times \left\lfloor \text{clamp}\left(\frac{x^r}{s}, -2^{n-1}, 2^{n-1} - 1\right) \right\rceil \quad (1)$$

where $\lfloor \cdot \rceil$ denotes rounding to the nearest integer, and $\text{clamp}(x, r_{\text{low}}, r_{\text{high}})$ constrains $x$ within the range $[r_{\text{low}}, r_{\text{high}}]$.

In neural networks with quantized weights and activations, the forward pass computes outputs using quantized values:

$$\text{Forward:} \quad \text{Output}(x) = x^q \cdot w^q = q(x^r) \cdot q(w^r) \quad (2)$$

Since the quantization function is non-differentiable, the Straight-Through Estimator (STE) (Bengio et al., 2013) is employed for gradient computation during backpropagation:

$$\text{Backward:} \quad \frac{\partial \mathcal{L}}{\partial x^r} = \begin{cases} \frac{\partial \mathcal{L}}{\partial x^q} & \text{if } x \in [-Q_N^x, Q_P^x] \\ 0 & \text{otherwise} \end{cases} \quad (3)$$

where $\mathcal{L}$ denotes the loss function, and $[-Q_N^x, Q_P^x]$ represents the quantization range. The gradient with respect to weights follows an analogous formulation.

For image classification tasks, the quantized model maps input images to predicted probability distributions through the forward function $f(x; w^q)$. Given a sample $(x_i, y_i)$ from the dataset $D$, the quantized model's output probability distribution is:

$$p_{\mathbf{Q}}(x_i; w^q) = \text{softmax}(f(x_i^q; w^q)) \quad (4)$$

For notational simplicity, we hereafter use $x$ to denote the quantized input and $\theta$ to represent the quantized model parameters unless otherwise specified.

The discrete nature of quantized parameters further amplifies the challenges identified above. First, the constrained parameter space (limited to $2^n$ possible values per weight) magnifies the systematic biases introduced by the "learning wrong answers" paradigm, as QNNs have limited capacity to absorb label noise. Second, the STE-based gradient approximation exacerbates directional conflicts between forgetting and retain gradients, making conflict-free optimization even more critical. These challenges motivate the design of our proposed OEU framework.

## 4. Methodology

In this section, we provide a comprehensive introduction to the proposed **OEU (Orthogonal Entropy Unlearning)**

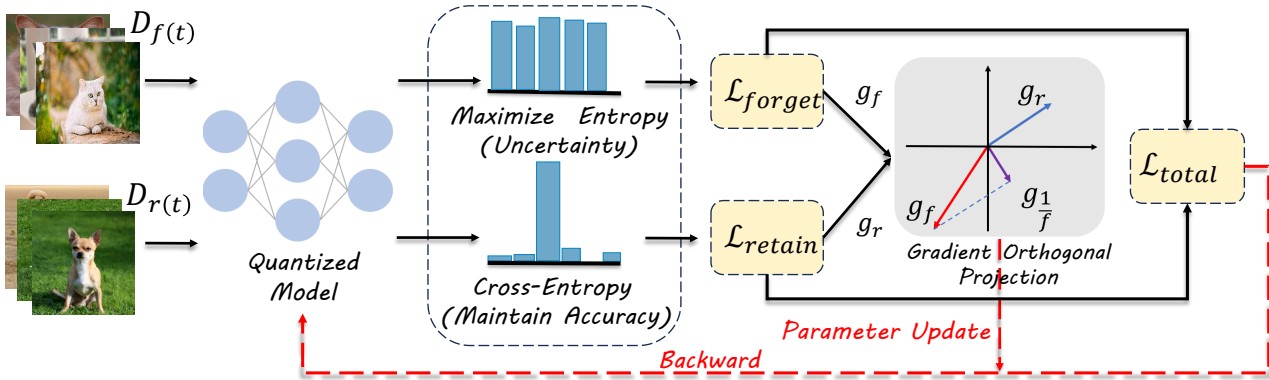

*Figure 2.* Overview of the proposed OEU framework. The forget set is trained with entropy maximization to achieve uncertainty, while the retain set uses cross-entropy to maintain accuracy. Gradient orthogonal projection resolves conflicts between the two gradients before parameter update.

framework. We first introduce the overview of OEU. Subsequently, we provide a detailed description of two core components of OEU: Entropy-Guided Unlearning (Section 4.2) and Gradient Orthogonal Projection (Section 4.3).

### 4.1. The Overview of OEU

In Figure 2, we present an overview of OEU, which consists of two parallel branches for processing the forget set and the retain set, followed by a gradient reconciliation module. Given a forget set $D_f$ and a retain set $D_r$, both are fed into the quantized model to obtain predictions. For the forget set, we apply entropy maximization to guide predictions away from confident outputs without bias toward any particular class, providing an unbiased forgetting direction rather than learning incorrect answers. For the retain set, we employ the standard cross-entropy loss to maintain prediction accuracy on retained data. The two objectives produce the forgetting gradient $g_f$ and the retain gradient $g_r$, respectively. To resolve potential directional conflicts between these gradients, we introduce a gradient orthogonal projection module that projects $g_f$ onto the orthogonal complement of $g_r$, ensuring that unlearning updates do not interfere with retained knowledge. The projected gradients are then combined to compute the total loss for parameter update. In the following two subsections, we will provide a detailed introduction to the two core components of OEU.

### 4.2. Entropy-Guided Unlearning

Existing unlearning methods, including Q-MUL's Similar Labels, operate under a "learning wrong answers" paradigm—they assign incorrect labels to forgotten samples and train the model to memorize these mislabels. We argue that this approach conflates forgetting with misremembering, potentially introducing systematic biases. Instead, we propose **Entropy-Guided Unlearning (EGU)** to render

the model genuinely uncertain about forgotten data.

For a quantized model $f_\theta^Q$ with parameters $\theta$, given an input $x$, the output probability distribution is:

$$p_\theta(k|x) = \frac{\exp(z_k)}{\sum_{j=1}^K \exp(z_j)}, \quad k \in \{1, 2, \ldots, K\}, \quad (5)$$

where $z = f_\theta^Q(x)$ denotes the logits and $K$ is the number of classes. The entropy of this distribution measures the model's uncertainty:

$$H(p_\theta(x)) = -\sum_{k=1}^K p_\theta(k|x) \log p_\theta(k|x), \quad (6)$$

The entropy is maximized when $p_\theta(k|x) = \frac{1}{K}$ for all $k$, yielding $H_{\max} = \log K$. This uniform distribution represents the theoretical optimum of $\mathcal{L}_{\text{forget}}$ in isolation. We provide a visualization of this effect in Appendix A.2.

We define the **entropy-guided forgetting loss** as the negative entropy over the forgotten dataset $D_f$:

$$\mathcal{L}_{\text{forget}} = -\mathbb{E}_{x \in D_f} \left[ H(p_\theta(x)) \right]$$
$$= \mathbb{E}_{x \in D_f} \left[ \sum_{k=1}^K p_\theta(k|x) \log p_\theta(k|x) \right]. \quad (7)$$

Minimizing $\mathcal{L}_{\text{forget}}$ maximizes the entropy, driving the model toward maximum uncertainty on forgotten samples. To preserve performance on retained data $D_r$, we employ the standard cross-entropy retain loss $\mathcal{L}_{\text{retain}} = \mathbb{E}_{(x,y) \in D_r} \left[ -\log p_\theta(y|x) \right]$.

**Theoretical Analysis of EGU.** We provide theoretical justifications for EGU by comparing it with label manipulation methods. Let $\mathcal{U}(k) = \frac{1}{K}$ denote the uniform distribution over $K$ classes, representing the ideal "forgotten" state where the model has no preference for any class.

**Theorem 4.1** (Bias of Label Manipulation). *Label manipulation methods (Random Labels, Similar Labels) minimize cross-entropy with incorrect labels $\tilde{y} \neq y$. The optimal prediction distribution is $p^*(k|x) = \mathbf{1}[k = \tilde{y}]$, yielding $H(p^*) = 0$ and $D_{KL}(p^* \| \mathcal{U}) = \log K$.*

**Theorem 4.2** (Unbiasedness of Maximum Entropy). *EGU maximizes $H(p_\theta(\cdot|x))$. The optimal prediction distribution is $p^*(k|x) = \frac{1}{K}$ for all $k$, yielding $H(p^*) = \log K$ and $D_{KL}(p^* \| \mathcal{U}) = 0$.*

Theorem 4.1 shows that label manipulation methods train the model to confidently predict incorrect labels, replacing correct memorization with incorrect memorization. This introduces systematic bias that may be exploited by membership inference attacks. In contrast, Theorem 4.2 shows that EGU provides an unbiased optimization direction toward higher entropy, avoiding class-specific biases inherent in label manipulation methods. Detailed proofs are provided in Appendix A.7.

### 4.3. Gradient Orthogonal Projection

While EGU defines an appropriate forgetting objective, naively optimizing it may conflict with the retain objective. Let $g_f = \nabla_\theta \mathcal{L}_{\text{forget}}$ and $g_r = \nabla_\theta \mathcal{L}_{\text{retain}}$ denote the gradients from forgetting and retain losses. When their cosine similarity $\cos(\theta_{fr}) = \frac{\langle g_f, g_r \rangle}{\|g_f\| \cdot \|g_r\|} < 0$, any update along $g_f$ will increase $\mathcal{L}_{\text{retain}}$. This conflict is particularly severe in quantized models, where the constrained parameter space amplifies gradient interference. Q-MUL's Adaptive Gradient Reweighting adjusts gradient magnitudes, but scalar weighting cannot resolve *directional* conflicts.

To eliminate such conflicts, we propose **Gradient Orthogonal Projection (GOP)**, which projects the forgetting gradient onto the orthogonal complement of the retain gradient.

$$g_f^\perp = g_f - \frac{\langle g_f, g_r \rangle}{\|g_r\|^2} g_r, \tag{8}$$

By construction, $\langle g_f^\perp, g_r \rangle = 0$, ensuring that updates along $g_f^\perp$ do not affect the retain loss under first-order approximation.

**Layer-wise Normalized Projection.** In quantized models, gradients propagated through the Straight-Through Estimator (STE) contain approximation errors that vary across layers. We apply layer-wise normalization before projection:

$$\tilde{g}_f^{(l)} = \frac{g_f^{(l)}}{\|g_f^{(l)}\| + \epsilon}, \quad \tilde{g}_r^{(l)} = \frac{g_r^{(l)}}{\|g_r^{(l)}\| + \epsilon}, \tag{9}$$

where $l$ indexes the layer and $\epsilon$ ensures numerical stability. The layer-wise orthogonal projection is:

$$g_f^{\perp(l)} = \tilde{g}_f^{(l)} - \langle \tilde{g}_f^{(l)}, \tilde{g}_r^{(l)} \rangle \cdot \tilde{g}_r^{(l)}, \quad \hat{g}_f^{(l)} = g_f^{\perp(l)} \cdot \|g_f^{(l)}\|. \tag{10}$$

We also introduce a hyperparameter $\alpha \in [0, 1]$ to control the degree of orthogonalization: $g_f^{(\alpha)} = g_f - \alpha \cdot \frac{\langle g_f, g_r \rangle}{\|g_r\|^2} g_r$. When $\alpha = 0$, no projection is applied; when $\alpha = 1$, full orthogonalization is enforced, allowing practitioners to trade off between forgetting aggressiveness and retain protection.

**Theoretical Analysis of GOP.** We provide theoretical justifications for GOP under first-order approximation. We assume $g_r \neq \mathbf{0}$, which holds when the retain set is non-trivial. The following theorem establishes that GOP guarantees conflict-free optimization.

**Theorem 4.3** (Retain Preservation). *Let $g_f^\perp = g_f - \frac{\langle g_f, g_r \rangle}{\|g_r\|^2} g_r$ be the projected forgetting gradient. For parameter update $\theta' = \theta - \eta g_f^\perp$, we have $\langle g_f^\perp, g_r \rangle = 0$, which implies $\mathcal{L}_r(\theta') \approx \mathcal{L}_r(\theta)$ under first-order approximation.*

**Theorem 4.4** (Forgetting Effectiveness). *If $g_f$ is not collinear with $g_r$, then $g_f^\perp \neq \mathbf{0}$ and $\langle g_f, g_f^\perp \rangle = \|g_f\|^2 \sin^2 \phi > 0$, where $\phi$ is the angle between $g_f$ and $g_r$. This guarantees $\mathcal{L}_f(\theta') < \mathcal{L}_f(\theta)$.*

Theorem 4.3 shows that the projected gradient is orthogonal to $g_r$, ensuring updates along $g_f^\perp$ do not increase retain loss. Theorem 4.4 confirms that projection preserves the descent direction for forgetting. Together, they guarantee that GOP achieves forgetting without harming retain. When gradients conflict ($\langle g_f, g_r \rangle < 0$), naive gradient descent increases $\mathcal{L}_r$, while GOP eliminates this interference entirely. Detailed proofs are provided in Appendix A.8.

### 4.4. Overall Pipeline

We now present the complete OEU framework that integrates entropy-guided unlearning with gradient orthogonal projection. The overall optimization objective combines the entropy-based forgetting loss and the cross-entropy retain loss:

$$\mathcal{L}_{\text{total}} = \mathcal{L}_{\text{forget}} + \beta \mathcal{L}_{\text{retain}}, \tag{11}$$

where $\beta$ balances the two objectives. However, rather than directly optimizing this combined loss, we compute separate gradients and apply orthogonal projection to ensure conflict-free updates. At each iteration, the model parameters are updated as:

$$\theta_{t+1} = \theta_t - \eta \left( \hat{g}_f^\perp + g_r \right). \tag{12}$$

where $\hat{g}_f^\perp$ is the orthogonally projected forgetting gradient and $g_r$ is the retain gradient. This update simultaneously drives the model toward uncertainty on forgotten data while preserving performance on retained data. The complete OEU algorithm is presented in Algorithm 1.

**Computational Complexity.** The computational overhead of OEU compared to standard unlearning is minimal.

---

**Algorithm 1** OEU: Orthogonal Entropy Unlearning

---

**Input:** Quantized model $\mathcal{M}_Q$ with parameters $\theta_0$, forgotten dataset $D_f$, retained dataset $D_r$, total epochs $T$, learning rate $\eta$, orthogonality strength $\alpha$

**Output:** Unlearned model $\mathcal{M}'_Q$ with parameters $\theta_u$

1: **for** $t \in [0, \ldots, T-1]$ **do**
2:     Compute entropy-guided forgetting loss $\mathcal{L}_{\text{forget}}$ by Eq. (7);
3:     Compute forgetting gradient $g_f \leftarrow \nabla_\theta \mathcal{L}_{\text{forget}}$;
4:     Compute retain loss $\mathcal{L}_{\text{retain}} \leftarrow \mathbb{E}_{(x,y) \in D_r} [\text{CE}(f_\theta(x), y)]$;
5:     Compute retain gradient $g_r \leftarrow \nabla_\theta \mathcal{L}_{\text{retain}}$;
6:     **for** each layer $l$ **do**
7:        Compute layer-wise gradients $\tilde{g}_f^{(l)}, \tilde{g}_r^{(l)}$ by Eq. (9);
8:        Compute orthogonal projection $\hat{g}_f^{(l)}$ by Eq. (10);
9:     **end for**
10:    Update parameters $\theta_{t+1} \leftarrow \theta_t - \eta \cdot (\hat{g}_f + g_r)$;
11: **end for**
12: **Return** Unlearned model $\mathcal{M}'_Q$ with parameters $\theta_T$

---

Entropy computation requires $O(K)$ per sample, where $K$ is the number of classes; GOP requires $O(|\theta|)$ per iteration, where $|\theta|$ is the number of parameters. Both operations are linear in their respective dimensions and negligible compared to the forward and backward passes. The overall time complexity remains $O(T \cdot (|D_f| + |D_r|) \cdot C_{\text{model}})$, where $C_{\text{model}}$ is the cost of a single forward-backward pass. We provide a runtime comparison of all methods in Appendix A.3.

**Synergy of Components.** The two components of OEU are complementary: *EGU* defines *what* to forget—providing an unbiased forgetting direction that avoids class-specific biases, while *GOP* and $\mathcal{L}_{\text{retain}}$ jointly constrain the forgetting degree to prevent over-unlearning; *GOP* defines *how* to forget—ensuring that the forgetting process does not interfere with retained knowledge and providing mathematical guarantees for utility preservation. Together, they form a principled framework for effective and conflict-free machine unlearning in QNNs.

## 5. Experiments

### 5.1. Experimental Setup

**Datasets and Networks.** Our experiments are conducted on four benchmark datasets: CIFAR-10 (Krizhevsky, 2009), CIFAR-100 (Krizhevsky, 2009), SVHN (Netzer et al., 2011), and Tiny-ImageNet (Le et al., 2015). Consistent with Q-MUL (Tong et al., 2025), we adopt ResNet-18 (He et al., 2016) with 4-bit weights and activations, and Mo-bileNetV2 (Howard et al., 2017) with 2-bit weights and full-precision activations. We evaluate under two unlearning scenarios: random data forgetting with forget ratios of 10%, 30%, and 50%, and class-wise forgetting. We primarily focus on random data forgetting as it better reflects real-world privacy removal requests.

**Baselines.** We compare OEU against eight methods. **Re-train** serves as the gold standard by retraining from scratch on retained data. **Fine-Tuning (FT)** (Golatkar et al., 2020; Warnecke et al., 2021) fine-tunes the original model on retained data. **Gradient Ascent (GA)** (Graves et al., 2021; Thudi et al., 2022) reverses learning by performing gradient ascent on forgotten data. **Influence Unlearning (IU)** (Koh & Liang, 2017; Izzo et al., 2021) removes data influence via Newton-step updates. Label manipulation methods include **Random Labels (RL)** (Golatkar et al., 2020), $\ell_1$-**sparse** (Jia et al., 2023) that combines random labeling with weight sparsification, and **SalUn** (Fan et al., 2024) that incorporates gradient-based saliency maps. **Q-MUL** (Tong et al., 2025) is the state-of-the-art method specifically designed for QNNs.

**Evaluation Metrics.** Following established protocols (Jia et al., 2023; Fan et al., 2024; Tong et al., 2025), we adopt five complementary metrics to assess both forgetting effectiveness and utility preservation. **Forget Accuracy (FA)** measures accuracy on forgotten data, which should ideally match that of a retrained model. **Retain Accuracy (RA)** evaluates accuracy on retained data to ensure knowledge preservation. **Test Accuracy (TA)** assesses generalization on held-out data. **Membership Inference Attack (MIA)** (Shokri et al., 2017) quantifies privacy leakage by measuring the distinguishability between forgotten and unseen samples. **Average Gap (AG)** computes the mean absolute difference between each method and Retrain across the above four metrics. Since Retrain represents the gold standard, effective unlearning is characterized by minimal gaps rather than simply higher or lower values. Therefore, AG serves as our primary indicator, where lower values indicate closer approximation to the ideal retrained model. We provide further elaboration in the Appendix A.1.

**Implementation details.** We adopt LSQ+ (Bhalgat et al., 2020) as the default quantization method. For training, we use SGD with an initial learning rate of 0.1 and cosine annealing. The original model is trained for 182 epochs, while Retrain uses 160 epochs with identical optimizer settings. We primarily focus on random data forgetting due to its practical relevance. Class-wise forgetting results are provided in the Appendix A.5, and additional training configurations, hyperparameter settings are presented in Appendix A.6

### 5.2. Performance Evaluation

As presented in Table 1, we evaluate various MU methods for quantized ResNet18 under random data forgetting. OEU consistently achieves the best performance across different forgetting ratios on both datasets. On CIFAR-10 with 10% forgetting ratio, OEU attains FA of 93.58% and RA of 99.86%, with minimal gaps of only 0.20% and 0.14% from Retrain, significantly outperforming Q-MUL which has gaps of 3.30% and 0.33%. The average gap of OEU

*Table 1.* Performance of various MU methods for ResNet18 with both activations and weights quantized to 4 bits CIFAR-10 and CIFAR-100. The unlearning scenario is random data forgetting. **Bold** indicates the best performance and underline indicates the runner-up. A performance gap against Retrain is provided in(•).

| Method | CIFAR-10 | | | | | CIFAR-100 | | | | |
|---|---|---|---|---|---|---|---|---|---|---|
| | FA | RA | TA | MIA | AG↓ | FA | RA | TA | MIA | AG↓ |
| The proportion of forgotten data samples to all samples is 10% | | | | | | | | | | |
| Retrain | 93.38 | 100.0 | 92.64 | 13.36 | 0 | 74.76 | 99.98 | 72.43 | 56.36 | 0 |
| FT | 99.42(6.04) | 99.96(0.04) | 93.05(0.41) | 2.82(10.54) | 4.26 | 91.82(17.06) | 94.12(5.86) | 65.29(7.14) | 16.69(39.67) | 17.43 |
| GA | 99.38(6.00) | 99.38(0.62) | 93.35(0.71) | 1.22(12.14) | 4.87 | 95.02(20.26) | 96.86(3.12) | 71.84(0.59) | 9.56(46.80) | 17.69 |
| IU | 95.64(2.26) | 95.53(4.47) | 87.85(4.79) | 7.93(5.43) | 4.24 | 3.69(71.07) | 4.03(95.95) | 3.85(68.58) | 98.8(42.44) | 69.51 |
| RL | 96.29(2.91) | 98.96(1.04) | 92.23(0.41) | 16.67(3.31) | 1.92 | 68.51(6.25) | 98.91(1.07) | 69.47(2.96) | 85.62(29.26) | 9.89 |
| $\ell_1$-sparse | 98.71(5.33) | 99.88(0.12) | 92.92(0.28) | 5.29(8.07) | 3.45 | 88.20(13.44) | 99.65(0.33) | 70.80(1.63) | 29.64(26.72) | 10.53 |
| SalUn | 96.82(3.44) | 99.45(0.55) | 92.32(0.32) | 14.27(0.91) | 1.31 | 82.22(7.46) | 98.71(1.27) | 67.38(5.05) | 66.78(10.42) | 6.05 |
| Q-MUL | 97.11(3.3) | 99.67(0.33) | 92.39(0.25) | 12.49(0.87) | 1.30 | 75.71(0.95) | 97.89(2.09) | 67.27(5.16) | 52.11(4.25) | 3.11 |
| OEU | 93.58(0.20) | 99.86(0.14) | 92.97(0.33) | 15.00(1.64) | **0.58** | 74.86(0.10) | 99.36(0.62) | 67.61(4.82) | 49.56(6.80) | **3.08** |
| The proportion of forgotten data samples to all samples is 30% | | | | | | | | | | |
| Retrain | 91.61 | 99.97 | 91.29 | 16.11 | 0 | 67.67 | 99.98 | 67.54 | 58.47 | 0 |
| FT | 99.19(7.58) | 99.95(0.02) | 92.94(1.65) | 3.10(13.01) | 5.57 | 96.24(28.57) | 99.94(0.04) | 71.49(3.95) | 20.57(37.90) | 17.62 |
| GA | 99.35(7.74) | 99.38(0.59) | 93.39(2.10) | 1.28(14.83) | 6.32 | 96.73(29.06) | 97.30(2.68) | 73.35(5.81) | 6.19(52.28) | 22.46 |
| IU | 76.21(15.40) | 75.93(24.04) | 71.48(19.81) | 24.53(8.42) | 16.92 | 2.81(64.86) | 3.19(96.79) | 3.09(64.45) | 96.33(37.86) | 65.99 |
| RL | 93.96(2.35) | 96.03(3.94) | 90.34(0.95) | 19.07(2.96) | 2.55 | 75.50(7.83) | 99.58(0.40) | 66.56(0.98) | 87.33(28.86) | 9.52 |
| $\ell_1$-sparse | 99.06(7.45) | 99.90(0.07) | 93.01(1.72) | 4.54(11.57) | 5.20 | 88.59(20.92) | 99.79(0.19) | 70.72(3.18) | 33.16(25.31) | 12.40 |
| SalUn | 96.17(4.56) | 97.78(2.19) | 91.57(0.30) | 15.50(0.66) | 1.93 | 79.60(11.93) | 96.70(3.28) | 63.41(4.13) | 57.73(0.74) | 5.02 |
| Q-MUL | 93.57(1.96) | 96.77(3.20) | 90.24(1.05) | 15.59(0.52) | 1.68 | 73.90(6.23) | 97.63(2.35) | 65.80(1.74) | 63.76(5.29) | 3.90 |
| OEU | 95.25(3.64) | 99.83(0.14) | 91.64(0.35) | 15.02(1.09) | **1.31** | 66.41(1.26) | 98.40(1.58) | 64.59(2.95) | 56.36(2.11) | **1.98** |
| The proportion of forgotten data samples to all samples is 50% | | | | | | | | | | |
| Retrain | 90.75 | 100.0 | 90.28 | 18.76 | 0 | 65.40 | 99.99 | 65.74 | 68.19 | 0 |
| FT | 99.29(8.54) | 99.96(0.04) | 93.28(3.00) | 2.85(15.91) | 6.87 | 97.10(31.70) | 99.98(0.01) | 72.82(7.08) | 16.86(51.33) | 22.71 |
| GA | 99.34(8.59) | 99.34(0.66) | 93.44(3.16) | 1.27(17.49) | 7.48 | 96.82(31.42) | 97.33(2.66) | 73.37(7.63) | 5.93(62.26) | 25.99 |
| IU | 16.12(74.63) | 16.02(83.98) | 16.22(74.06) | 16.36(2.40) | 58.77 | 0.96(64.44) | 1.04(98.95) | 1.00(64.74) | 98.96(30.77) | 64.73 |
| RL | 95.32(4.57) | 97.17(2.83) | 90.85(0.57) | 21.33(2.57) | 2.64 | 69.20(3.80) | 85.60(14.39) | 64.39(1.35) | 54.58(13.61) | 8.29 |
| $\ell_1$-sparse | 99.25(8.50) | 99.95(0.05) | 92.67(2.39) | 3.91(14.85) | 6.45 | 96.48(31.08) | 99.98(0.01) | 72.26(6.52) | 25.11(43.08) | 20.17 |
| SalUn | 96.56(5.81) | 98.03(1.97) | 91.55(1.27) | 18.05(0.71) | 2.44 | 87.75(22.35) | 98.99(1.00) | 64.69(1.05) | 73.12(4.93) | 7.33 |
| Q-MUL | 91.09(0.34) | 93.94(6.06) | 89.12(1.16) | 18.68(0.08) | 1.91 | 67.25(1.85) | 89.65(10.34) | 61.38(4.36) | 56.52(11.67) | 7.06 |
| OEU | 95.25(4.50) | 99.64(0.36) | 90.81(0.59) | 18.72(0.04) | **1.37** | 70.42(5.02) | 98.79(1.20) | 60.38(5.36) | 65.08(3.11) | **3.67** |

*Table 2.* Impact of Removing Components of OEU. Ablation study is from ResNet18 on CIFAR-100. The unlearning scenario is random data forgetting (30%).

| Method | FA | RA | TA | MIA | AG↓ |
|---|---|---|---|---|---|
| Retrain | 67.67 | 99.98 | 67.54 | 58.47 | 0 |
| OEU | 66.41(1.26) | 98.40(1.58) | 64.59(2.95) | 56.36(2.11) | **1.98** |
| *w/o*GOP | 63.16(4.51) | 97.13(2.85) | 63.87(3.67) | 61.54 (3.07) | 3.53 |
| *w/o*EGU | 75.10(7.43) | 99.43(0.55) | 67.64(0.10) | 50.62(7.85) | 3.98 |
| $GOP_{global}$ | 57.99(9.68) | 97.04(2.94) | 62.35(5.19) | 62.24(3.77) | 5.40 |

is only 0.58%, which is 55% lower than Q-MUL's 1.30%. As the forgetting ratio increases to 30% and 50%, OEU maintains its superiority with average gaps of 1.31% and 1.37%, compared to Q-MUL's 1.68% and 1.91% respectively. These results demonstrate that OEU effectively removes the influence of forgotten data while preserving the model's knowledge on retained data.

On CIFAR-100, the advantages of OEU become even more pronounced. At 10% forgetting ratio, OEU achieves an average gap of 3.08%, slightly better than Q-MUL's 3.11%. When the forgetting ratio increases to 30%, OEU achieves FA of 66.41% with a gap of 1.26% from Retrain, resulting in an average gap of only 1.98%, approximately 49% lower than Q-MUL's 3.90%. This indicates that OEU maintains su-

perior unlearning effectiveness even under more challenging conditions. At 50% forgetting ratio, OEU achieves an average gap of 3.67%, substantially outperforming Q-MUL's 7.06% and SalUn's 7.33%. These results validate the effectiveness of OEU in balancing forgetting efficacy and model utility preservation for QNNs. More additional results can be found in the Appendix A.4 and A.9.

### 5.3. The Impact of Different Components

We conduct ablation experiments with two variants: *w/o* GOP and *w/o* EGU. As shown in Table 3, the complete OEU achieves an average gap of only 1.98%. When GOP is removed, AG rises to 3.53%, with the RA gap increasing from 1.58% to 2.85% and the FA gap from 1.26% to 4.51%. This demonstrates that without orthogonal projection, the forgetting gradient conflicts with the retain gradient, degrading both forgetting precision and retain performance. When EGU is removed, the model reverts to the "learning wrong labels" paradigm, resulting in an FA of 75.10%—significantly higher than the retrained model's 67.67%—indicating incomplete forgetting. The MIA gap also increases from 2.11% to 7.85%, revealing weaker privacy protection. These results validate that both components are indispensable: EGU defines the appropriate forgetting

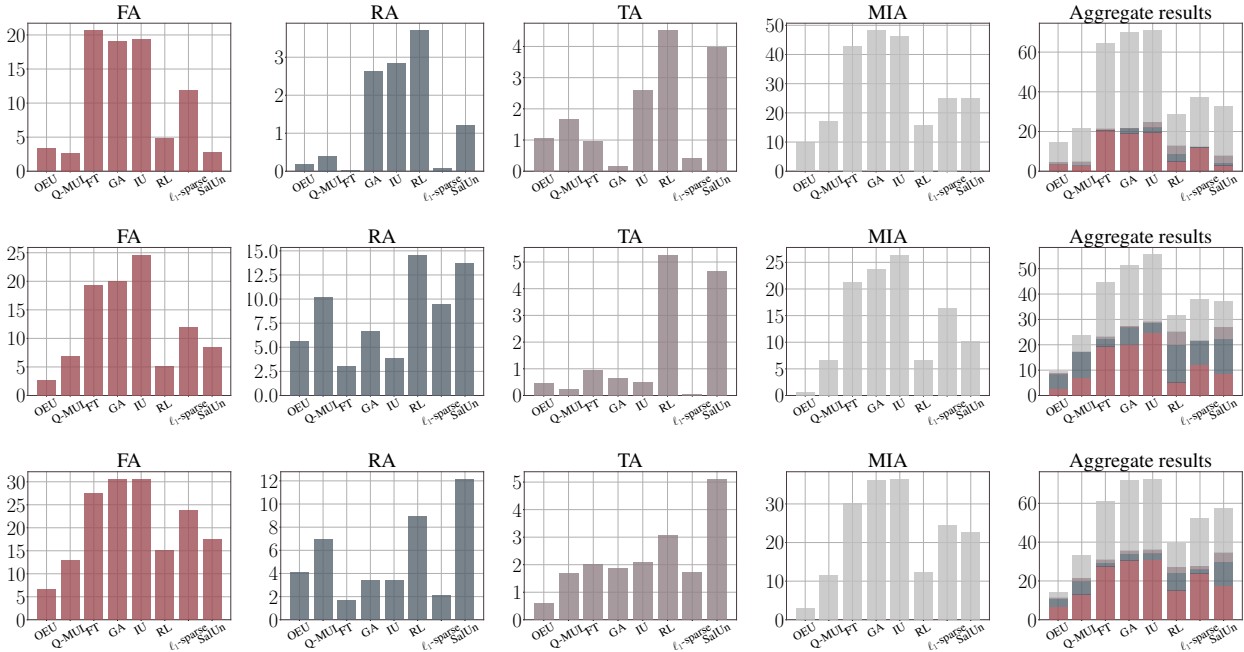

*Figure 3.* Performance gaps of ResNet18 on CIFAR-100. The three rows correspond to full-precision ResNet18, ResNet18 quantized with PACT (Choi et al., 2018), and ResNet18 quantized with DSQ (Gong et al., 2019), respectively. The unlearning scenario is random data forgetting (10%). A shorter bar (smaller gap) indicates performance is closer to that of the Retrained model. The average gap for each method is calculated by dividing the values in the Aggregate Results bar chart by 4.

objective, while GOP ensures conflict-free optimization.

We further investigate the impact of projection granularity by comparing layer-wise projection with global projection. Replacing layer-wise GOP with global projection increases AG from 1.98% to 5.40%, with FA dropping from 66.41% to 57.99%, far below the retrained model's 67.67%, indicating over-forgetting. Global projection computes a single orthogonal direction across all parameters, failing to capture layer-specific gradient dynamics. In contrast, layer-wise projection allows each layer to independently resolve gradient conflicts, providing finer-grained control over the forgetting-retain trade-off.

### 5.4. OEU with Full-precision Models

To more comprehensively evaluate the generalizability of OEU beyond quantized settings, we compare the effects of different unlearning methods on full-precision models, as shown in Figure 3. The issues of directional gradient conflicts and biased forgetting objectives also exist in full-precision models, although they are not as severe as in quantized models with constrained parameter spaces. Nevertheless, OEU can still effectively address these two issues. Specifically, the average gap of OEU reaches only 3.60%, which is significantly lower than that of existing methods. This demonstrates that OEU can achieve superior unlearning performance even in full-precision models, further validating the effectiveness of our approach.

### 5.5. OEU with Different QAT Methods

Beyond LSQ+, we validate OEU under other QAT methods including PACT (Choi et al., 2018) and DSQ (Gong et al., 2019). As shown in Figure 3, OEU consistently achieves the smallest performance gaps across all metrics for both quantization methods. Compared to Q-MUL, OEU reduces the average gap by 60.5% for PACT and 56.2% for DSQ. Moreover, OEU demonstrates superior MIA performance, indicating more thorough forgetting of target data. These results validate the generalizability of OEU across different quantization-aware training frameworks.

## 6. Conclusions

In this paper, we address machine unlearning for quantized neural networks by identifying two fundamental limitations of existing methods: the "learning wrong answers" paradigm that introduces biases, and scalar gradient reweighting that ignores directional conflicts. We propose orthogonal entropy unlearning, featuring entropy-guided unlearning that provides an unbiased forgetting direction by maximizing prediction uncertainty, and gradient orthogonal projection that mathematically guarantees conflict-free optimization. Extensive experiments demonstrate that OEU consistently outperforms existing methods across multiple datasets and quantization configurations.

## Impact Statement

This paper presents work whose goal is to advance the field of machine learning, with direct relevance to privacy regulations such as GDPR that mandate individuals' right to data removal. Incomplete forgetting and residual information leakage are shared risks across all approximate unlearning methods. OEU specifically mitigates these: entropy-guided unlearning drives predictions toward maximum uncertainty, inherently reducing identifiable patterns from forgotten data; gradient orthogonal projection ensures forgetting does not compromise retained knowledge, maintaining a stable and reliable unlearned model.

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

# A. Appendix

## A.1. More Discussion on Evaluation Metrics

We provide additional justification for our choice of Average Gap (AG) as the primary evaluation metric.

**Consistency with Existing Literature.** We adopt AG as the primary evaluation metric to maintain consistency with the established machine unlearning literature. Baseline methods such as $\ell_1$-sparse (Jia et al., 2023), SalUn (Fan et al., 2024),and Q-MUL (Tong et al., 2025) employ the same evaluation framework, which facilitates direct and objective comparisons across different approaches.

**Measuring Deviation from the Gold Standard.** The fundamental purpose of AG is to quantify the overall deviation of an unlearning method from the "gold standard"—the Retrain baseline, which represents the ideal unlearning outcome achieved by retraining from scratch exclusively on the retain set $D_r$. By computing the mean absolute deviation across all four metrics, AG provides a holistic assessment of how closely an approximate unlearning method approximates this ideal behavior.

**Balanced Evaluation of Dual Objectives.** Machine unlearning inherently involves two competing objectives: (1) *unlearning effectiveness*—ensuring that the influence of forgotten data is thoroughly removed, and (2) *utility preservation*—maintaining the model's performance on retained and unseen data. Our evaluation framework captures both objectives through a balanced 2:2 design:

- **Unlearning Effectiveness:** Forget Accuracy (FA) and Membership Inference Attack success rate (MIA) jointly evaluate how effectively the model has forgotten the target data.

- **Utility Preservation:** Retain Accuracy (RA) and Test Accuracy (TA) assess the model's utility on retained training data and held-out test data, respectively. These metrics ensure that the unlearning process does not catastrophically degrade the model's predictive capability.

The equal-weight averaging in AG thus reflects a principled balance between these dual demands, rather than biasing toward any single aspect. This design choice ensures that methods achieving strong unlearning at the cost of utility degradation (or vice versa) are appropriately penalized, promoting the development of approaches that excel across both dimensions.

In summary, the AG metric provides a comprehensive, balanced, and comparable evaluation framework that aligns with the fundamental goals of machine unlearning research.

## A.2. Visualization of Entropy-Guided Unlearning

To provide qualitative evidence for the effectiveness of our entropy-guided unlearning objective, we visualize the output probability distribution of a randomly selected sample from the forget set. The experiment is conducted on CIFAR-100 using MobileNetV2 with 2-bit weight quantization. As shown in Figure 4, the original model produces a peaked distribution with high confidence on a single class (left). After applying our entropy-guided unlearning, the output transforms into a nearly uniform distribution (right), where each class receives approximately equal probability. This visualization shows the isolated effect of entropy maximization. In the complete OEU pipeline, joint optimization with $\mathcal{L}_{\text{retain}}$ constrains the entropy increase through shared features, preventing over-unlearning while avoiding incorrect memorization.

## A.3. Efficiency Analysis

We compare the runtime of different unlearning methods. As illustrated in Figure 5, Retrain incurs substantial computational cost, requiring approximately 20 minutes to complete. While GA and IU exhibit minimal time overhead, their poor unlearning performance on quantized models (as evidenced in Table 1) renders them impractical. OEU maintains comparable efficiency to other approximate methods such as Q-MUL, SalUn, and RL, completing within approximately 3 minutes despite incorporating entropy maximization and gradient orthogonal projection. This is because both operations are computationally lightweight—entropy computation is $\mathcal{O}(K)$ per sample and layer-wise projection is dominated by standard backpropagation. Combined with its superior unlearning effectiveness demonstrated in our experiments, OEU achieves a favorable balance between performance and computational efficiency.

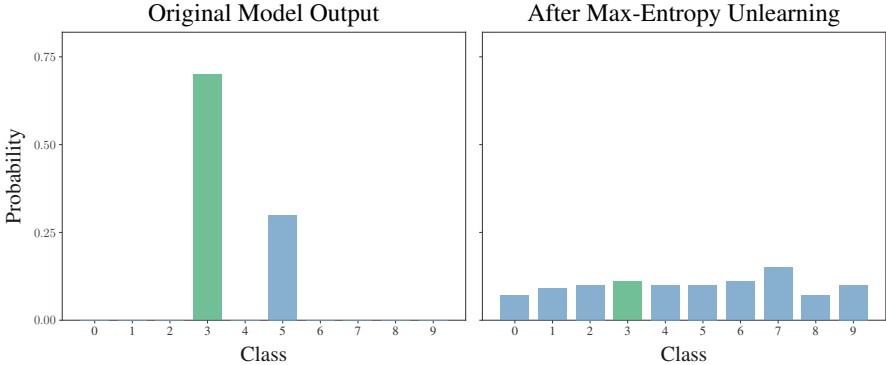

*Figure 4.* Output probability distribution of a randomly selected forget sample from CIFAR-100 on MobileNetV2 (green: ground-truth class; blue: other classes). Left: output distribution of original model . Right: output distribution after applying max-entropy unlearning.

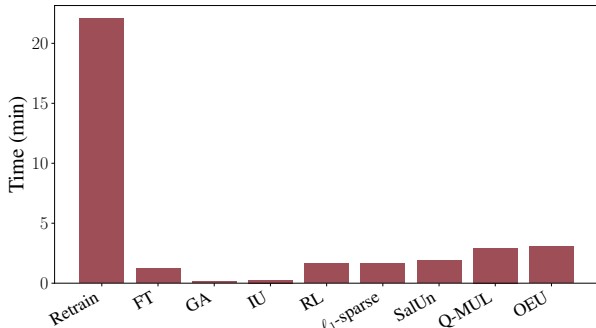

*Figure 5.* Runtime comparison of different unlearning methods on MobileNetV2 with CIFAR-100 under 10% random data forgetting. OEU achieves comparable efficiency to other approximate methods while delivering superior unlearning performance.

### A.4. Performance Evaluation on Tiny-Imagenet dataset

To validate the scalability of OEU on larger datasets, we conduct experiments on Tiny-ImageNet, which contains 200 classes with higher image resolution. As shown in Table 3, OEU achieves the best average gap of only 5.03%, significantly outperforming Q-MUL (8.17%) and $\ell_1$-sparse (9.02%). Notably, OEU attains an FA of 60.84%, closely matching the Retrain baseline (62.15%) with a gap of merely 1.31%, and achieves the smallest MIA gap of 3.39%, indicating superior privacy protection. These results demonstrate that OEU maintains its effectiveness on larger and more complex datasets, further validating its practical applicability.

### A.5. Performance Evaluation under Class-wise Forgetting

We further evaluate OEU under the class-wise forgetting scenario, where all samples from a specific class are removed. As shown in Table 4, on CIFAR-10, most approximate methods achieve near-optimal performance, with $\ell_1$-sparse (0.08%), SalUn (0.10%), Q-MUL (0.11%), and OEU (0.12%) all attaining average gaps below 0.15%. This is because class-wise forgetting on CIFAR-10 (10 classes) is a relatively simple task where multiple methods can effectively eliminate the target class. However, on the more challenging CIFAR-100 dataset with 100 classes (Table 5), OEU demonstrates clear advantages, achieving the best average gap of 0.21%, outperforming Q-MUL (0.32%) and $\ell_1$-sparse (0.55%). OEU perfectly forgets the target class (FA=0.00%) while maintaining minimal degradation on RA (0.34% gap) and TA (0.51% gap). These results indicate that OEU scales effectively to more complex class-wise forgetting scenarios.

*Table 3.* Performance of various MU methods for ResNet18 with 4-bit quantization on Tiny-Imagenet. The unlearning scenario is random data forgetting(10%). **Bold** indicates the best performance and underline indicates the runner-up.

| Method | Tiny-Imagenet | | | | |
|---|---|---|---|---|---|
| | FA | RA | TA | MIA | AG↓ |
| Retrain | 62.15 | 99.46 | 62.65 | 51.51 | 0 |
| FT | 74.45(12.30) | 91.99(7.47) | 56.57(6.08) | 33.67(17.84) | 10.92 |
| GA | 92.95(30.8) | 93.01(6.45) | 57.19(5.46) | 11.77(39.74) | 20.61 |
| IU | 0.45(61.70) | 0.51(98.95) | 0.50(62.15) | 0.45(51.06) | 68.47 |
| RL | 60.05(2.10) | 82.60(16.86) | 51.59(11.06) | 45.13(6.38) | 9.10 |
| $\ell_1$-sparse | 60.89(1.26) | 75.44(24.02) | 56.41(6.24) | 46.94(4.57) | 9.02 |
| SalUn | 65.28(3.13) | 81.70(17.76) | 51.77(10.88) | 37.08(14.43) | 11.55 |
| Q-MUL | 72.10(9.95) | 91.99(7.47) | 56.53(6.32) | 42.56(8.95) | **8.17** |
| OEU | 60.84(1.31) | 92.69(6.77) | 54.01(8.64) | 48.12(3.39) | **5.03** |

*Table 4.* Performance of various MU methods for ResNet18 with 4-bit quantization on CIFAR-10. The unlearning scenario is class-wise forgetting (one class is forgotten.). **Bold** indicates the best performance and underline indicates the runner-up.

| Method | CIFAR-10 | | | | |
|---|---|---|---|---|---|
| | FA | RA | TA | MIA | AG↓ |
| Retrain | 0.00 | 99.99 | 93.34 | 100.0 | 0 |
| FT | 62.40(62.40) | 99.95(0.04) | 93.59(0.25) | 94.67(5.33) | 17.02 |
| GA | 11.20(11.20) | 92.28(7.71) | 85.49(7.85) | 92.13(7.87) | 8.66 |
| IU | 0.00(0.00) | 29.75(70.24) | 28.92(64.42) | 100.0(0.00) | 33.67 |
| RL | 0.00(0.00) | 99.87(0.12) | 93.74(0.40) | 100.0(0.00) | 0.13 |
| $\ell_1$-sparse | 0.00(0.00) | 99.85(0.14) | 93.51(0.17) | 100.0(0.00) | **0.08** |
| SalUn | 0.04(0.04) | 99.84(0.15) | 93.56(0.22) | 100.0(0.00) | 0.10 |
| Q-MUL | 0.00(0.00) | 99.69(0.30) | 93.20(0.13) | 100.0(0.00) | 0.11 |
| OEU | 0.00(0.00) | 99.82(0.17) | 93.64(0.30) | 100.0(0.00) | 0.12 |

*Table 5.* Performance of various MU methods for ResNet18 with 4-bit quantization on CIFAR-100. The unlearning scenario is class-wise forgetting (one class is forgotten). **Bold** indicates the best performance and underline indicates the runner-up.

| Method | CIFAR-100 | | | | |
|---|---|---|---|---|---|
| | FA | RA | TA | MIA | AG↓ |
| Retrain | 0.00 | 99.96 | 70.63 | 100.0 | 0 |
| FT | 7.11(7.11) | 99.79(0.27) | 71.10(0.47) | 100.0(0.00) | 1.96 |
| GA | 37.56(37.56) | 88.69(11.27) | 61.75(8.88) | 79.78(20.22) | 19.48 |
| IU | 0.00(0.00) | 72.25(17.71) | 51.22(19.41) | 100.0(0.00) | 9.28 |
| RL | 0.89(0.89) | 99.97(0.01) | 72.78(2.15) | 100.0(0.00) | 0.76 |
| $\ell_1$-sparse | 0.00(0.00) | 98.33(1.63) | 70.06(0.57) | 100.0(0.00) | 0.55 |
| SalUn | 1.33(1.33) | 99.92(0.04) | 73.12(2.49) | 100.0(0.00) | 0.97 |
| Q-MUL | 0.00(0.00) | 99.50(0.46) | 69.79(0.84) | 100.0(0.00) | 0.32 |
| OEU | 0.00(0.00) | 99.62(0.34) | 70.12(0.51) | 100.0(0.00) | **0.21** |

## A.6. Hyperparameter Settings and Training Details

We follow the experimental settings of SalUn and Q-MUL for baseline methods, and all experiments use the SGD optimizer on a single NVIDIA RTX 4090 GPU. For FT and RL, we train for 10 epochs with learning rates searched within [1e-3, 1e-1]. For GA, we train for 5 epochs with learning rates in [1e-5, 1e-3]. For IU, we search the parameter $\alpha$ associated with the Woodfisher Hessian inverse approximation within [1, 20]. For $\ell_1$-sparse, we search the parameter $\gamma$ within [1e-6, 1e-4] and learning rates within [1e-3, 1e-1]. For SalUn, we train for 10 epochs with learning rates in [5e-4, 5e-2] and sparsity

ratios in [0.1, 0.9]. For Q-MUL, we train for 10 epochs with learning rates in the range [1e-3, 1e-1]. For our proposed OEU, we train for 10 epochs with learning rates searched within [1e-2, 1e-1] and a batch size of 256, with the orthogonality strength $\alpha$ set to 1.0 by default across all datasets and forgetting scenarios.

## A.7. Theoretical Analysis of Entropy-Guided Unlearning

This section provides theoretical justifications for entropy-guided unlearning by analyzing its properties compared to label manipulation methods.

### A.7.1. PROBLEM SETUP

Consider a $K$-class classification problem. For a forget sample $(x, y)$ with true label $y \in \{1, \ldots, K\}$, the model produces a prediction distribution $p_\theta(\cdot|x) = \text{softmax}(f_\theta(x))$. We define the uniform distribution $\mathcal{U}(k) = \frac{1}{K}$ as the reference distribution representing the ideal "forgotten" state, where the model has no knowledge about the sample.

### A.7.2. ANALYSIS OF LABEL MANIPULATION METHODS

Label manipulation methods assign incorrect labels to forget samples and train the model via cross-entropy loss.

**Definition A.1** (Random Labels). For a forget sample $(x, y)$, Random Labels assigns a random incorrect label $\tilde{y} \sim \text{Uniform}(\{1, \ldots, K\} \setminus \{y\})$ and minimizes:

$$\mathcal{L}_{RL}(\theta; x, \tilde{y}) = -\log p_\theta(\tilde{y}|x) \tag{13}$$

**Definition A.2** (Similar Labels). Similar Labels assigns the most similar incorrect label $\tilde{y} = \arg\max_{k \neq y} \text{sim}(k, y)$ and minimizes the same cross-entropy loss.

*Proof of Theorem 4.1.* The cross-entropy loss $\mathcal{L}_{RL}(\theta; x, \tilde{y}) = -\log p_\theta(\tilde{y}|x)$ is minimized when $p_\theta(\tilde{y}|x) = 1$. Under the constraint $\sum_{k=1}^{K} p_\theta(k|x) = 1$, the optimal distribution is:

$$p^*(k|x) = \mathbf{1}[k = \tilde{y}] = \begin{cases} 1 & k = \tilde{y} \\ 0 & k \neq \tilde{y} \end{cases} \tag{14}$$

The entropy of this distribution is:

$$H(p^*) = -\sum_{k=1}^{K} p^*(k|x) \log p^*(k|x) = -1 \cdot \log 1 = 0 \tag{15}$$

The KL divergence from the uniform distribution is:

$$D_{KL}(p^*\|\mathcal{U}) = \sum_{k=1}^{K} p^*(k) \log \frac{p^*(k)}{\mathcal{U}(k)}$$
$$= 1 \cdot \log \frac{1}{1/K} = \log K \tag{16}$$

$\square$

*Remark* A.3. The optimal solution $p^*(k|x) = \mathbf{1}[k = \tilde{y}]$ indicates that label manipulation methods train the model to memorize incorrect labels with full confidence. This does not achieve genuine forgetting but rather replaces correct memorization with incorrect memorization.

### A.7.3. ANALYSIS OF ENTROPY-GUIDED UNLEARNING

Entropy-guided unlearning maximizes the prediction entropy on forget samples.

**Definition A.4** (Entropy-Guided Unlearning). For a forget sample $x$, entropy-guided unlearning minimizes the negative entropy:

$$\mathcal{L}_{ME}(\theta; x) = -H(p_\theta(\cdot|x)) = \sum_{k=1}^{K} p_\theta(k|x) \log p_\theta(k|x) \tag{17}$$

*Proof of Theorem 4.2.* We seek to maximize entropy $H(p) = -\sum_{k=1}^{K} p_k \log p_k$ subject to $\sum_{k=1}^{K} p_k = 1$ and $p_k \geq 0$. Using the method of Lagrange multipliers:

$$\mathcal{L}(p, \lambda) = -\sum_{k=1}^{K} p_k \log p_k + \lambda \left( \sum_{k=1}^{K} p_k - 1 \right) \tag{18}$$

Taking the derivative with respect to $p_k$ and setting it to zero:

$$\frac{\partial \mathcal{L}}{\partial p_k} = -\log p_k - 1 + \lambda = 0 \tag{19}$$

This yields $p_k = e^{\lambda-1}$, which is constant for all $k$. Applying the constraint $\sum_{k=1}^{K} p_k = 1$:

$$K \cdot e^{\lambda-1} = 1 \implies e^{\lambda-1} = \frac{1}{K} \tag{20}$$

Therefore, the optimal distribution is:

$$p^*(k|x) = \frac{1}{K}, \quad \forall k \in \{1, \ldots, K\} \tag{21}$$

The entropy of the uniform distribution is:

$$H(p^*) = -\sum_{k=1}^{K} \frac{1}{K} \log \frac{1}{K} = \log K \tag{22}$$

The KL divergence from the uniform distribution is:

$$D_{KL}(p^* \| \mathcal{U}) = \sum_{k=1}^{K} \frac{1}{K} \log \frac{1/K}{1/K} = 0 \tag{23}$$

$\square$

### A.7.4. EXPECTED BIAS ANALYSIS

We further analyze the expected behavior of Random Labels when considering the randomness in label assignment.

**Proposition A.5** (Expected Target Distribution). *Let $\tilde{y}$ be sampled uniformly from $\{1, \ldots, K\} \setminus \{y\}$. The expected target distribution induced by Random Labels is:*

$$\bar{p}(k|x) = \mathbb{E}_{\tilde{y}}[\mathbf{1}[k = \tilde{y}]] = \begin{cases} 0 & k = y \\ \frac{1}{K-1} & k \neq y \end{cases} \tag{24}$$

*Proof.* For $k = y$, since $\tilde{y} \neq y$ by construction, we have $\mathbf{1}[k = \tilde{y}] = 0$ with probability 1.

For $k \neq y$, the probability that $\tilde{y} = k$ is $\frac{1}{K-1}$ since $\tilde{y}$ is sampled uniformly from the $K-1$ incorrect labels. $\square$

**Proposition A.6** (Bias of Expected Distribution). *The expected target distribution $\bar{p}$ has non-zero divergence from the uniform distribution:*

$$D_{KL}(\bar{p} \| \mathcal{U}) = \log \frac{K}{K-1} > 0 \tag{25}$$

*Proof.* Computing the KL divergence:

$$D_{KL}(\bar{p}\|\mathcal{U}) = \sum_{k=1}^{K} \bar{p}(k) \log \frac{\bar{p}(k)}{\mathcal{U}(k)}$$

$$= \bar{p}(y) \log \frac{\bar{p}(y)}{\mathcal{U}(y)} + \sum_{k \neq y} \bar{p}(k) \log \frac{\bar{p}(k)}{\mathcal{U}(k)} \tag{26}$$

For $k = y$: $\bar{p}(y) = 0$, and by convention $0 \cdot \log 0 = 0$, so this term contributes 0.

For $k \neq y$ (total of $K - 1$ terms):

$$\sum_{k \neq y} \frac{1}{K-1} \log \frac{1/(K-1)}{1/K} = (K-1) \cdot \frac{1}{K-1} \cdot \log \frac{K}{K-1}$$

$$= \log \frac{K}{K-1} \tag{27}$$

Since $K \geq 2$, we have $\frac{K}{K-1} > 1$, thus $D_{KL}(\bar{p}\|\mathcal{U}) > 0$. $\qquad\square$

*Remark* A.7. Proposition A.6 reveals that even in expectation, Random Labels introduces a systematic bias: the true class $y$ is suppressed to probability 0 rather than the uniform $\frac{1}{K}$. This asymmetry may be exploited by membership inference attacks to distinguish forgotten samples from unseen samples.

## A.8. Theoretical Analysis of Gradient Orthogonal Projection

This section provides detailed proofs for the theoretical results presented in Section 4. We establish that Gradient Orthogonal Projection (GOP) guarantees conflict-free optimization under first-order approximation.

### A.8.1. PRELIMINARIES AND ASSUMPTIONS

Consider a quantized neural network with parameters $\theta \in \mathbb{R}^d$. Let $\mathcal{L}_f(\theta)$ and $\mathcal{L}_r(\theta)$ denote the forgetting and retain losses, respectively, with corresponding gradients:

$$g_f = \nabla_\theta \mathcal{L}_f(\theta), \quad g_r = \nabla_\theta \mathcal{L}_r(\theta) \tag{28}$$

**Assumption A.8.** The retain gradient is non-vanishing, i.e., $g_r \neq \mathbf{0}$. This assumption holds in practice when the retain set contains sufficient samples for meaningful gradient computation.

The projected forgetting gradient is defined as:

$$g_f^\perp = g_f - \frac{\langle g_f, g_r \rangle}{\|g_r\|^2} g_r \tag{29}$$

### A.8.2. PROOF OF THEOREM 4.3 (RETAIN PRESERVATION)

We first establish a key lemma showing that the projected gradient is orthogonal to the retain gradient.

**Lemma A.9** (Orthogonality). *The projected gradient $g_f^\perp$ satisfies $\langle g_f^\perp, g_r \rangle = 0$.*

*Proof.* By the definition of $g_f^\perp$ in Eq. (29):

$$\langle g_f^\perp, g_r \rangle = \left\langle g_f - \frac{\langle g_f, g_r \rangle}{\|g_r\|^2} g_r, \, g_r \right\rangle$$

$$= \langle g_f, g_r \rangle - \frac{\langle g_f, g_r \rangle}{\|g_r\|^2} \langle g_r, g_r \rangle$$

$$= \langle g_f, g_r \rangle - \frac{\langle g_f, g_r \rangle}{\|g_r\|^2} \cdot \|g_r\|^2$$

$$= \langle g_f, g_r \rangle - \langle g_f, g_r \rangle = 0 \tag{30}$$

$\square$

*Proof of Theorem 4.3.* The orthogonality $\langle g_f^\perp, g_r \rangle = 0$ follows directly from Lemma A.9. To show retain preservation, we apply Taylor's first-order expansion of $\mathcal{L}_r$ at $\theta$:

$$
\begin{aligned}
\mathcal{L}_r(\theta') &= \mathcal{L}_r(\theta - \eta g_f^\perp) \\
&\approx \mathcal{L}_r(\theta) + \langle \nabla_\theta \mathcal{L}_r(\theta), -\eta g_f^\perp \rangle \\
&= \mathcal{L}_r(\theta) - \eta \langle g_r, g_f^\perp \rangle \\
&= \mathcal{L}_r(\theta) - \eta \cdot 0 = \mathcal{L}_r(\theta)
\end{aligned}
\tag{31}
$$

Thus, the retain loss remains unchanged under first-order approximation. $\square$

### A.8.3. PROOF OF THEOREM 4.4 (FORGETTING EFFECTIVENESS)

*Proof.* We prove the two claims sequentially.

**Claim 1: $g_f^\perp \neq 0$ when $g_f$ is not collinear with $g_r$.**

Suppose for contradiction that $g_f^\perp = 0$. By Eq. (29):

$$
g_f = \frac{\langle g_f, g_r \rangle}{\|g_r\|^2} g_r = \lambda g_r
\tag{32}
$$

where $\lambda = \frac{\langle g_f, g_r \rangle}{\|g_r\|^2}$. This implies $g_f$ is collinear with $g_r$, contradicting the assumption. Therefore, $g_f^\perp \neq 0$.

**Claim 2: $\langle g_f, g_f^\perp \rangle = \|g_f\|^2 \sin^2 \phi > 0$.**

Expanding the inner product:

$$
\begin{aligned}
\langle g_f, g_f^\perp \rangle &= \left\langle g_f, g_f - \frac{\langle g_f, g_r \rangle}{\|g_r\|^2} g_r \right\rangle \\
&= \|g_f\|^2 - \frac{\langle g_f, g_r \rangle^2}{\|g_r\|^2}
\end{aligned}
\tag{33}
$$

Let $\phi$ denote the angle between $g_f$ and $g_r$. By the definition of cosine:

$$
\cos \phi = \frac{\langle g_f, g_r \rangle}{\|g_f\| \|g_r\|}
\tag{34}
$$

Substituting:

$$
\begin{aligned}
\langle g_f, g_f^\perp \rangle &= \|g_f\|^2 - \frac{\|g_f\|^2 \|g_r\|^2 \cos^2 \phi}{\|g_r\|^2} \\
&= \|g_f\|^2 (1 - \cos^2 \phi) \\
&= \|g_f\|^2 \sin^2 \phi
\end{aligned}
\tag{35}
$$

Since $g_f$ and $g_r$ are non-collinear, $\phi \notin \{0, \pi\}$, implying $\sin \phi \neq 0$. Combined with $g_f \neq 0$, we have $\langle g_f, g_f^\perp \rangle > 0$.

**Descent guarantee.** Applying Taylor's expansion to $\mathcal{L}_f$:

$$
\mathcal{L}_f(\theta - \eta g_f^\perp) \approx \mathcal{L}_f(\theta) - \eta \langle g_f, g_f^\perp \rangle < \mathcal{L}_f(\theta)
\tag{36}
$$

for $\eta > 0$, since $\langle g_f, g_f^\perp \rangle > 0$. $\square$

## A.9. Performance evaluation on MobileNetV2

To further validate the generalizability of OEU, we conduct experiments on MobileNetV2 with 2-bit weight quantization across CIFAR-10, CIFAR-100, and SVHN datasets under random data forgetting scenarios. As shown in Tables 6, 7, and 8, OEU consistently achieves the lowest average gap across all datasets and forgetting ratios, demonstrating its effectiveness on lightweight architectures with aggressive quantization.

*Table 6.* Performance of various MU methods for MobileNetV2 with activations kept at full precision and weights quantized to 2 bits on CIFAR-10. The unlearning scenario is random data forgetting. **Bold** indicates the best performance and underline indicates the runner-up. A performance gap against Retrain is provided in (•).

| Method | CIFAR-10 | | | | |
|---|---|---|---|---|---|
| | FA | RA | TA | MIA | AG↓ |
| The proportion of forgotten data samples to all samples is 10% | | | | | |
| Retrain | 85.97 | 94.39 | 85.49 | 18.60 | 0 |
| FT | 93.64 (7.67) | 95.42 (1.03) | 87.37 (1.88) | 9.62 (8.98) | 4.89 |
| GA | 93.97 (8.00) | 94.87 (0.48) | 87.21 (1.72) | 8.49 (10.11) | 5.08 |
| IU | 13.13(72.84) | 14.28(80.11) | 13.97(71.52) | 84.51(66.91) | 75.85 |
| RL | 85.82 (0.15) | 87.46 (6.93) | 84.05 (1.44) | 17.27 (1.33) | 2.46 |
| $\ell_1$-sparse | 93.17 (7.20) | 95.15 (0.76) | 87.66 (2.17) | 9.89 (8.71) | 4.71 |
| SalUn | 90.87(4.90) | 92.08(2.31) | 86.63(1.14) | 15.71(2.89) | 2.81 |
| Q-MUL | 90.71(4.74) | 93.60(0.79) | 87.51(2.02) | 15.00(3.60) | 2.79 |
| OEU | 88.00(2.03) | 91.86(2.53) | 85.98(0.49) | 16.78(1.82) | **1.72** |

| Method | CIFAR-10 | | | | |
|---|---|---|---|---|---|
| | FA | RA | TA | MIA | AG↓ |
| The proportion of forgotten data samples to all samples is 30% | | | | | |
| Retrain | 84.87 | 93.87 | 84.19 | 19.90 | 0 |
| FT | 91.53 (6.66) | 93.17 (0.70) | 86.21 (2.02) | 12.07 (7.83) | 4.30 |
| GA | 89.16 (4.29) | 89.40 (4.47) | 82.79 (1.40) | 15.20 (4.70) | 3.72 |
| IU | 12.67(72.20) | 13.11(80.76) | 12.65(71.54) | 86.47(66.57) | 72.77 |
| RL | 88.04 (3.17) | 88.86 (5.01) | 84.26 (0.07) | 21.81(1.91) | 2.54 |
| $\ell_1$-sparse | 92.90 (8.03) | 94.53 (0.66) | 86.80 (2.61) | 10.76 (9.14) | 5.11 |
| SalUn | 95.20(10.33) | 96.83(2.96) | 90.49(6.30) | 16.87(3.03) | 5.66 |
| Q-MUL | 87.56(2.69) | 89.24(4.63) | 85.63(1.44) | 18.56(1.34) | 2.53 |
| OEU | 86.75(1.88) | 89.93(3.94) | 82.94(1.25) | 20.27(0.37) | **1.86** |

| Method | CIFAR-10 | | | | |
|---|---|---|---|---|---|
| | FA | RA | TA | MIA | AG↓ |
| The proportion of forgotten data samples to all samples is 50% | | | | | |
| Retrain | 82.27 | 94.76 | 82.17 | 22.17 | 0 |
| FT | 89.50 (7.23) | 91.26 (3.50) | 84.32 (2.15) | 13.05 (9.12) | 5.5 |
| GA | 75.99 (6.28) | 76.07 (18.69) | 72.07 (10.10) | 24.30 (2.13) | 9.3 |
| IU | 21.89(60.38) | 22.03(72.73) | 21.67(60.50) | 78.52(56.35) | 62.49 |
| RL | 86.81 (4.54) | 87.85 (6.91) | 83.70(1.53) | 17.98 (4.19) | 4.29 |
| $\ell_1$-sparse | 6.64 (11.09) | 95.70 (0.94) | 87.11 (4.94) | 10.00 (12.17) | 7.29 |
| SalUn | 87.98(5.71) | 88.95(5.81) | 84.56(2.39) | 15.04(7.13) | 5.26 |
| Q-MUL | 87.08(4.81) | 89.48(5.28) | 84.76(2.59) | 22.24(0.07) | 3.19 |
| OEU | 87.62(5.35) | 90.77(3.99) | 83.37(1.20) | 21.33(0.84) | **2.85** |

*Table 7.* Performance of various MU methods for MobileNetV2 with activations kept at full precision and weights quantized to 2 bits on CIFAR-100. The unlearning scenario is random data forgetting. **Bold** indicates the best performance and underline indicates the runner-up. A performance gap against Retrain is provided in (•).

| Method | CIFAR-100 | | | | |
| --- | --- | --- | --- | --- | --- |
| | FA | RA | TA | MIA | AG↓ |
| The proportion of forgotten data samples to all samples is 10% | | | | | |
| Retrain | 61.53 | 89.44 | 60.90 | 40.20 | 0 |
| FT | 82.89 (21.36) | 88.18 (1.26) | 63.56 (2.66) | 19.11 (21.09) | 11.59 |
| GA | 83.82 (22.29) | 85.20 (4.24) | 61.80 (0.90) | 17.33 (22.87) | 12.58 |
| IU | 2.40(59.13) | 2.81(86.63) | 2.76(58.14) | 2.62(37.58) | 60.37 |
| RL | 81.53 (20.00) | 88.41 (1.03) | 63.26 (2.36) | 26.38 (13.82) | 9.30 |
| $\ell_1$-sparse | 81.31 (19.78) | 85.97 (3.47) | 62.33 (1.43) | 19.98 (20.22) | 11.23 |
| SalUn | 82.22(20.69) | 88.15(1.29) | 63.26(2.36) | 27.38(12.82) | 9.29 |
| Q-MUL | 69.96(8.43) | 80.25(9.19) | 62.29(1.39) | 33.15(7.05) | 6.52 |
| OEU | 69.29(7.76) | 85.79(3.65) | 61.72(0.82) | 39.27(0.93) | **3.29** |

| Method | CIFAR-100 | | | | |
| --- | --- | --- | --- | --- | --- |
| | FA | RA | TA | MIA | AG↓ |
| The proportion of forgotten data samples to all samples is 30% | | | | | |
| Retrain | 53.90 | 83.38 | 54.97 | 46.46 | 0 |
| FT | 79.36 (25.46) | 86.34 (2.96) | 62.04 (7.07) | 22.03 (24.43) | 14.98 |
| GA | 66.02 (12.12) | 66.34 (17.04) | 51.44 (3.53) | 24.67 (21.79) | 13.62 |
| IU | 6.68(47.22) | 7.30(76.08) | 6.75(48.22) | 94.62(48.16) | 54.92 |
| RL | 77.16 (23.26) | 83.49(0.11) | 61.31 (6.34) | 25.43(21.03) | 12.69 |
| $\ell_1$-sparse | 82.53 (28.63) | 88.77 (5.39) | 63.30 (8.33) | 19.75 (26.71) | 17.27 |
| SalUn | 76.26(22.36) | 83.46(0.08) | 61.19(6.22) | 26.08(20.38) | 12.26 |
| Q-MUL | 67.77(13.87) | 79.60(3.78) | 61.99(7.02) | 34.87(11.59) | 9.07 |
| OEU | 65.59(11.69) | 78.09(5.29) | 55.68(0.71) | 51.55(5.09) | **5.70** |

| Method | CIFAR-100 | | | | |
| --- | --- | --- | --- | --- | --- |
| | FA | RA | TA | MIA | AG↓ |
| The proportion of forgotten data samples to all samples is 50% | | | | | |
| Retrain | 49.20 | 85.03 | 50.21 | 51.44 | 0 |
| FT | 83.76 (34.56) | 91.24 (6.21) | 63.92 (13.71) | 20.10(31.34) | 18.65 |
| GA | 41.20 (8.00) | 40.60 (44.43) | 33.94 (16.27) | 46.70 (5.44) | 18.54 |
| IU | 1.40(47.80) | 1.51(83.52) | 1.26(48.95) | 22(29.44) | 52.43 |
| RL | 75.36 (26.16) | 80.92 (4.11) | 60.57(10.36) | 33.59(17.85) | 14.62 |
| $\ell_1$-sparse | 18.69 (19.78) | 85.97 (3.47) | 62.33 (1.43) | 19.98 (20.22) | 11.23 |
| SalUn | 73.56(24.36) | 79.50(5.53) | 59.31(9.10) | 34.78(16.66) | 13.91 |
| Q-MUL | 67.93(18.73) | 78.83(6.20) | 60.38(10.17) | 44.41(7.03) | 10.53 |
| OEU | 64.41(15.21) | 75.56(9.47) | 52.80(2.59) | 47.50(3.94) | **7.80** |

*Table 8.* Performance of various MU methods for MobileNetV2 with activations kept at full precision and weights quantized to 2 bits on SVHN. The unlearning scenario is random data forgetting. **Bold** indicates the best performance and underline indicates the runner-up. A performance gap against Retrain is provided in (•).

| Method | SVHN | | | | |
|---|---|---|---|---|---|
| | FA | RA | TA | MIA | AG↓ |
| *The proportion of forgotten data samples to all samples is 10%* | | | | | |
| Retrain | 94.25 | 99.99 | 94.17 | 9.33 | 0 |
| FT | 99.12 (4.87) | 99.98 (0.01) | 94.71 (0.54) | 2.18 (7.15) | 3.14 |
| GA | 99.21 (4.96) | 99.43 (0.56) | 94.84 (0.67) | 10.77 (1.44) | 1.91 |
| IU | 98.65 (4.40) | 98.83 (1.16) | 93.96 (0.21) | 2.43 (6.90) | 3.17 |
| RL | 96.81 (2.56) | 98.36 (1.63) | 94.46 (0.29) | 20.20 (10.87) | 3.84 |
| $\ell_1$-sparse | 99.07 (4.82) | 99.97 (0.02) | 94.67 (0.50) | 2.68 (6.65) | 3.00 |
| SalUn | 96.42 (2.17) | 98.19 (1.80) | 94.51 (0.34) | 20.39 (11.06) | 3.84 |
| Q-MUL | 96.47 (2.22) | 99.48 (0.51) | 94.96 (0.79) | 9.16 (0.17) | 0.92 |
| OEU | 94.40 (0.15) | 99.82 (0.17) | 94.25 (0.08) | 11.17 (1.84) | **0.56** |

| Method | SVHN | | | | |
|---|---|---|---|---|---|
| | FA | RA | TA | MIA | AG↓ |
| *The proportion of forgotten data samples to all samples is 30%* | | | | | |
| Retrain | 93.12 | 100.0 | 93.81 | 11.21 | 0 |
| FT | 99.28 (6.16) | 99.98 (0.02) | 94.72 (0.91) | 2.13 (9.08) | 4.04 |
| GA | 99.27 (6.15) | 99.46 (0.54) | 94.77 (0.96) | 1.08 (10.13) | 4.45 |
| IU | 98.63 (5.41) | 98.83 (1.17) | 93.05 (0.76) | 2.92 (8.29) | 3.91 |
| RL | 94.55 (1.43) | 96.99 (3.01) | 93.79 (0.02) | 22.33 (11.12) | 3.90 |
| $\ell_1$-sparse | 99.21 (6.09) | 99.96 (0.04) | 94.81 (1.00) | 2.53 (8.68) | 3.95 |
| SalUn | 95.33 (2.21) | 96.75 (3.25) | 93.88 (0.07) | 25.44 (14.23) | 4.94 |
| Q-MUL | 96.80 (3.68) | 99.46 (0.54) | 94.95 (1.14) | 12.77 (1.56) | 1.73 |
| OEU | 95.83 (2.71) | 99.83 (0.17) | 93.94 (0.13) | 9.66 (1.55) | **1.14** |

| Method | SVHN | | | | |
|---|---|---|---|---|---|
| | FA | RA | TA | MIA | AG↓ |
| *The proportion of forgotten data samples to all samples is 50%* | | | | | |
| Retrain | 92.57 | 95.83 | 93.19 | 12.46 | 0 |
| FT | 99.26 (6.69) | 99.98 (4.15) | 94.59 (1.40) | 2.03 (10.43) | 5.67 |
| GA | 99.31 (6.74) | 99.48 (3.65) | 94.75 (1.56) | 10.53 (1.93) | 3.47 |
| IU | 97.82 (5.25) | 97.87 (2.04) | 92.90 (0.29) | 3.92 (8.54) | 4.03 |
| RL | 94.23 (1.66) | 95.36 (0.47) | 93.05 (0.14) | 32.43 (19.97) | 5.56 |
| $\ell_1$-sparse | 99.27 (6.70) | 99.97 (4.14) | 94.57 (1.38) | 2.44 (10.02) | 5.56 |
| SalUn | 93.93 (1.36) | 95.02 (0.81) | 92.87 (0.32) | 33.78 (21.32) | 5.95 |
| Q-MUL | 95.30 (2.73) | 98.50 (2.67) | 94.31 (1.12) | 19.90 (7.44) | 3.49 |
| OEU | 95.33 (2.76) | 99.79 (3.96) | 93.45 (0.26) | 12.18 (0.28) | **1.82** |

