# OpenReview forum: "Forget by Uncertainty: Orthogonal Entropy Unlearning for Quantized Neural Networks"
_ICML.cc/2026/Conference — ICML 2026 regular_

### Official Review · Reviewer_PUb2 · 2026-03-09

**Soundness:** 4
**Presentation:** 3
**Significance:** 3
**Originality:** 2
**Overall Recommendation:** 4
**Confidence:** 3

**Summary:**

The paper “Forget by Uncertainty: Orthogonal Entropy Unlearning for Quantized Neural Networks” deals with machine unlearning in quantized neural networks (QNNs). Real-world deployments use QNNs mostly, and gradient-based unlearning procedures, especially training with forget data assigned with random labels, incurs issues. This is because the weights are constrained, and any training update might jump quantization bins and cause large and inconsistent gradient updates. Only one other work ‘Q-MUL’ has explored unlearning in QNNs, and they reduce this random label noise by assigning the forget to similar labels. This paper claims that Q-MUL combines forgetting with remembering incorrectly and mis-biases towards specific classes and is therefore not genuine data removal from the model. Also, Q-MUL reweights the forget and retain gradients so that quantized models forget but not drift away from the retained data. Authors state Q-MUL does not consider the directionality of the gradient update, and any forget gradient which is not orthogonal to the retain gradient might be harmful. Their method “OEU, a novel Orthogonal Entropy Unlearning framework” challenges the narrative of mislabelling to forget, and instead drives the unlearning objective to drive towards an uniform distribution of labels instead of one particular label, and they project the gradients orthogonally to remove any gradient interference. OEU performs extensive evaluation on CIFAR-10, CIFAR-100 and Tiny-ImageNet for ResNet-18 network mostly. It achieves best results among SOTA works, and shows experiments on different kinds of quantization. Hence, OEU shows good analysis and hypothesis for machine unlearning objective on QNNs.

**Compliance With Llm Reviewing Policy:**

Affirmed.

**Final Justification:**

Authors have shown their method OEU achieves lowest error in 8-bit quantized non-residual CNNs and transformer architectures as per my queries, on CIFAR-100 datasets. They have also demonstrated multi-class removal for 10-30 classes, with better forgetting ability than SOTA on 4-bit quantized ResNets. They check both gradient-based and black-box attacks in the rebuttal and have answered all of our questions. The paper was already suitable for acceptance, the rebuttal reinforced it. Hence, I would like to maintain my score of weak accept.

**Key Questions For Authors:**

1)	Does the method work well with other DNNs like non-Residual nets and ViTs?
2)	What happens when multiple classes are removed?
3)	What happens in case of adversarial attacks (gradient-based and black-box)?

**Limitations:**

The authors have not specifically discussed the limitations in the paper, but they have discussed the empirical results including efficiency of their method in details.

**Strengths And Weaknesses:**

Strengths:
1) OEU correctly estimates the drawbacks in the current unlearning objective for QNN unlearning mechanisms, and shows that in their empirical results.
2) The experimentation is comprehensive: evaluation on CIFAR-10, CIFAR-100 and TinyImageNet on ResNet18 for random data and class wise forgetting and component removal ablations.
3) The experimentation includes different kinds of quantization- 4bit, DSQ and PACT, and their aggregate results favor OEU.
4) The paper is well-motivated and well-executed.

Weaknesses:
1) The authors do not show what happens when multiple classes are removed. It would be useful to also show results on networks other than ResNet, like ViTs, VGGs etc.
2) The paper utilizes the principles of orthogonal gradient projection for lessening forget and retain interference. The authors need to cite the papers where the basis for orthogonality of gradients come from: Gradient projection memory for continual learning, Gradient Surgery for Multi-Task learning etc.
3) Minor: The introduction could better explain what is traditionally done in MU for mislabelling type unlearning before attacking the flaws. Also, what do the authors mean by systematic biases for Q-MUL.

---

> ### Author Rebuttal · Authors · 2026-03-31
>
> **For W1 and Q1，Q2:**  We evaluate OEU on DeiT-Tiny and VGG-16 with 8-bit quantization on CIFAR-100 under 30% random data forgetting. OEU achieves the lowest AG on both architectures: 2.41% on DeiT-T versus 2.79% for Q-MUL and 4.43% for SalUn, and 4.08% on VGG versus 5.84% for Q-MUL and 6.16% for SalUn. These results confirm that OEU generalizes well to both transformer-based and deeper CNN architectures.
>
> DeiT-T on CIFAR-100 (30%)  :
>
> |         | FA    | RA    | TA    | MIA   | AG   |
> | ------- | ----- | ----- | ----- | ----- | ---- |
> | retrain | 81.43 | 93.58 | 81.09 | 20.40 | 0    |
> | SalUn   | 91.35 | 96.27 | 82.64 | 23.96 | 4.43 |
> | QMUL    | 82.64 | 98.02 | 82.93 | 24.08 | 2.79 |
> | OEU     | 82.75 | 97.56 | 82.71 | 23.11 | 2.41 |
>
> VGG-16 on CIFAR-100 (30%)  :
>
> |         | FA    | RA    | TA    | MIA   | AG   |
> | ------- | ----- | ----- | ----- | ----- | ---- |
> | retrain | 66.30 | 99.92 | 67.27 | 47.33 | 0    |
> | SalUn   | 84.24 | 99.21 | 70.75 | 44.81 | 6.16 |
> | QMUL    | 79.52 | 98.73 | 71.09 | 52.46 | 5.84 |
> | OEU     | 74.67 | 99.03 | 68.51 | 41.51 | 4.08 |
>
> **Multi-Class Forgetting.** We extend class-wise forgetting to removing 10 and 30 classes on CIFAR-100 using 4-bit ResNet18. When forgetting 10 classes, OEU achieves an AG of 0.56%, outperforming Q-MUL at 0.68% and SalUn at 0.92%. OEU validates its scalability to larger-scale class removal.
>
> Forget 10 classes:
>
> |         | FA   | RA    | TA    | MIA    | AG   |
> | ------- | ---- | ----- | ----- | ------ | ---- |
> | retrain | 0    | 99.97 | 73.60 | 100.00 | 0    |
> | SalUn   | 0.81 | 99.89 | 70.82 | 100.00 | 0.92 |
> | QMUL    | 0    | 99.23 | 71.64 | 100.00 | 0.68 |
> | OEU     | 0    | 99.38 | 71.96 | 100.00 | 0.56 |
>
> Forget 30 classes:
>
> |         | FA   | RA    | TA    | MIA    | AG   |
> | ------- | ---- | ----- | ----- | ------ | ---- |
> | retrain | 0    | 99.97 | 75.33 | 100.00 | 0    |
> | SalUn   | 1.06 | 99.51 | 71.89 | 100.00 | 1.24 |
> | QMUL    | 0    | 99.13 | 72.84 | 100.00 | 0.83 |
> | OEU     | 0    | 99.01 | 73.10 | 100.00 | 0.80 |
>
> **For W2:**  We will cite "Gradient Projection Memory for Continual Learning" [1] and "Gradient Surgery for Multi-Task Learning" [2] in Section 4.3.  We will also clarify our specific adaptations for the unlearning setting: (1) adapting gradient projection to machine unlearning, where forgetting and retaining are inherently opposing objectives, unlike the multi-task or continual learning scenarios in prior work; (2) introducing layer-wise normalized projection to handle STE-induced gradient noise in quantized networks; (3) integrating gradient projection with entropy-guided unlearning to form a unified framework for QNNs.
>
> [1] Saha, G., Garg, I., & Roy, K.(2021). Gradient Projection Memory for Continual Learning. In International Conference on Learning Representations.
>
> [2] Yu, T., Kumar, S., Gupta, A., Levine, S., Hausman, K., & Finn, C. (2020). Gradient surgery for multi-task learning. Advances in neural information processing systems, 33, 5824-5836.
>
> **For W3:** We will expand the introduction to explain how traditional MU mislabelling methods work: replacing true labels with incorrect ones and retraining via cross-entropy, steering the model away from correct predictions. Regarding "systematic biases" in Q-MUL: Similar Labels redirects confidence toward a fixed predetermined incorrect class (the most similar class to the true label). As shown in Figure 1(a), rather than achieving genuine unlearning with uncertainty, the model simply memorizes a specific wrong answer with high confidence, essentially replacing correct memorization with incorrect memorization. This means the influence of the forget data is not truly removed but merely redirected.
>
> **For Q3:** We evaluate adversarial robustness under both gradient-based attacks (FGSM, PGD-20) and black-box attack (Square Attack). The results show that OEU maintains adversarial robustness closely aligned with the Retrain baseline across all attack types. Specifically, OEU achieves 18.62% under Square Attack versus 19.08% for Retrain, and 4.72% under PGD-20 versus 3.78% for Retrain, with differences within a narrow margin. This indicates OEU does not introduce additional adversarial vulnerability.
>
> | Method  | TA(%) | FGSM(%) | PGD-20(%) | Square Attack(%) |
> | ------- | ----- | ------- | --------- | ---------------- |
> | Retrain | 69.67 | 17.32   | 3.78      | 19.08            |
> | SalUn   | 68.25 | 15.13   | 4.60      | 17.31            |
> | Q-MUL   | 69.56 | 16.91   | 7.48      | 18.24            |
> | OEU     | 69.40 | 16.39   | 4.72      | 18.62            |
>
> **For Limitations:** We will add a Limitations section discussing: (1) Current experiments focus on CNN-based architectures; our rebuttal results on DeiT confirm generalizability, and extending to larger-scale models remains future work. (2) Theoretical guarantees rely on first-order approximation, a standard assumption well-supported by our extensive empirical results.

---

> > ### Author Rebuttal · Reviewer_PUb2 · 2026-04-01
> >
> > We appreciate the explanations provided to our queries, and would like to maintain our score.

---

> > > ### Author Response · Authors · 2026-04-07
> > >
> > > Thank you for your feedback and for taking the time to review our rebuttal. We are glad to hear that your concerns have been addressed, and we truly appreciate your continued support.

---

### Official Review · Reviewer_AvH7 · 2026-03-10

**Soundness:** 4
**Presentation:** 4
**Significance:** 3
**Originality:** 3
**Overall Recommendation:** 4
**Confidence:** 3

**Summary:**

The paper studies the problem of machine unlearning in quantized neural networks. The authors identify two limitations of existing approaches: (1) label-perturbation-based methods transform correct predictions into deterministic incorrect predictions, which may introduce systematic bias rather than removing the influence of the forgotten data; and (2) adaptive gradient reweighting only rescales gradient magnitudes and cannot resolve directional conflicts between the forgetting and retaining gradients.

To address these issues, the paper proposes a method consisting of two components. First, it introduces an entropy-guided objective that increases predictive uncertainty on the forget samples. Second, it applies a gradient orthogonal projection mechanism to mitigate conflicts between forgetting and retaining updates.

The paper provides theoretical analysis for the proposed components and presents experimental results on multiple datasets and quantization settings, showing improvements over existing approaches across several evaluation metrics.

**Compliance With Llm Reviewing Policy:**

Affirmed.

**Final Justification:**

I appreciate the authors' efforts in addressing my comments. I will maintain my score for weak accept.

**Key Questions For Authors:**

1. The introduction states that machine unlearning becomes more challenging in quantized neural networks (QNNs). However, some of the issues discussed (e.g., bias introduced by incorrect labels and the directional conflict between forget and retain gradients) may also arise in full-precision models. It would therefore be helpful if the authors could further clarify what new technical challenges arise specifically from the combination of quantization and machine unlearning. In particular, which difficulties are intrinsic to quantized models, and which are general machine unlearning issues that become amplified in the quantized setting.

2. The experimental coverage is generally sufficient. However, in some settings the improvement of OEU over Q-MUL appears relatively small (e.g., for certain AG values). It would be helpful if the authors could clarify whether the reported results are averaged over multiple random seeds or different forget splits, and provide variability statistics where appropriate to better demonstrate the robustness of the improvements.

3. The orthogonality strength parameter α is used to balance the trade-off between forgetting and retaining performance. In the current experiments, α is fixed to 1.0. It would be helpful if the authors could provide a brief sensitivity analysis on this parameter to better demonstrate the robustness and stability of the proposed method.

**Limitations:**

While the authors include an impact statement, the discussion of limitations and potential societal impacts remains relatively brief. The discussion could be further strengthened by briefly commenting on how the method may behave under more challenging settings, such as large-scale data removal or stronger approximation. In addition, since the work relates to machine unlearning, it would be valuable to acknowledge potential risks such as incomplete unlearning or residual information leakage.

**Strengths And Weaknesses:**

Soundness: The paper analyzes two limitations of Q-MUL and proposes targeted improvements through entropy-guided forgetting and gradient orthogonal projection. The formulations of the proposed modules are clearly presented, and the paper provides supporting derivations together with empirical validation across multiple datasets and quantization settings. The ablation studies further help illustrate the contribution of each component.

Presentation: The paper is generally well organized, with a clear structure and logical progression. The motivation, method description, and experimental setup are presented in a reasonably clear manner, and the figures and tables help illustrate the proposed approach and results.

Significance: The paper studies machine unlearning in quantized neural networks, a direction that has received relatively limited attention in the current literature. By addressing both the formulation of the forgetting objective and the issue of gradient conflicts under quantization constraints, the work explores important aspects of this problem. The proposed method demonstrates improved performance over prior approaches across several experimental settings.

Originality: The paper introduces an entropy-guided forgetting objective together with a gradient orthogonal projection mechanism to improve existing QNN unlearning methods. While these components build on ideas explored in related areas, their integration for quantized model unlearning provides a reasonable methodological refinement for this relatively underexplored setting.

---

> ### Author Rebuttal · Authors · 2026-03-31
>
> **For Q1:**  We would like to clarify that the two challenges we identified — biased forgetting objectives and directional gradient conflicts — are indeed general MU issues, and we do not claim them to be exclusive to QNNs. In fact, we demonstrate in Sec 5.4 that OEU also achieves superior performance on full-precision models, precisely because these issues exist there as well. However, these challenges are significantly amplified in quantized models due to two intrinsic properties:
>
> (1) Constrained parameter space. Quantized models are restricted to discrete values, severely limiting the model's capacity to absorb conflicting optimization signals. Full-precision models' continuous parameter space provides sufficient degrees of freedom to partially accommodate both forgetting and retain objectives. QNNs lack this flexibility, making gradient conflicts more damaging. This is evidenced by certain methods' catastrophic failure—e.g., IU achieves AG of 69.51% on CIFAR-100 (Table 1), essentially producing a collapsed model, whereas such complete failures are less common in full-precision settings.
>
> (2) STE-induced gradient approximation errors. QNNs rely on the Straight-Through Estimator for backpropagation, which introduces layer-varying gradient approximation errors that exacerbate the directional mismatch between forgetting and retain gradients. This directly motivates our layer-wise normalized projection design, specifically tailored for this quantization-specific challenge. We will further clarify this distinction in the revised introduction.
>
> **For Q2:**  Regarding the experimental setup, all reported results are averaged over 5 random seeds, where each seed determines a different forget split. Importantly, all methods share the same 5 forget splits to ensure fair comparison. We will add standard deviations in the revised manuscript:
>
> |                       | FA         | RA         | TA         | MIA        | AG   |
> | --------------------- | ---------- | ---------- | ---------- | ---------- | ---- |
> | OEU on Cifar-10(30%)  | 95.25±0.45 | 99.83±0.13 | 91.64±0.37 | 15.02±0.79 | 1.31 |
> | OEU on Cifar-100(30%) | 66.41±0.94 | 98.40±0.21 | 64.59±0.80 | 56.36±0.86 | 1.98 |
>
> Regarding the magnitude of improvements, we would like to clarify that in settings where the AG gap between OEU and Q-MUL appears relatively small (e.g., CIFAR-100 at 10%: OEU 3.08% vs. Q-MUL 3.11%), both methods are already close to the Retrain gold standard, leaving inherently limited room for improvement. Importantly, as the forgetting ratio increases and the task becomes more challenging, OEU's advantage becomes substantially more pronounced — for example, on CIFAR-100 at 30%, OEU achieves AG of 1.98% vs. Q-MUL's 3.90% (49% relative reduction), and at 50%, 3.67% vs. 7.06% (48% relative reduction). This trend demonstrates that OEU's advantages are most evident in practically demanding scenarios.
>
> **For Q3:**  We have conducted sensitivity analysis for α on CIFAR-100 with ResNet18 (4-bit, 30% random forgetting). As α increases from 0 to 1, AG steadily decreases from 3.53% to 1.98%. Performance is stable across α=0.5 to 1.0, with AG varying only between 1.98% and 2.26%, demonstrating robustness. However, α=2 causes severe degradation (AG=13.62%) due to over-correction beyond orthogonality. Thus α=1.0 achieves the best balance and is adopted as default.
>
> |         | FA    | RA    | TA    | MIA   | AG    |
> | ------- | ----- | ----- | ----- | ----- | ----- |
> | retrain | 67.67 | 99.98 | 67.54 | 58.47 | 0     |
> | α=0 | 63.16 | 97.13 | 63.87 | 61.54 | 3.53  |
> | α=0.1 | 63.24 | 97.26 | 63.26 | 61.21 | 3.12  |
> | α=0.3 | 63.68 | 98.01 | 64.21 | 60.68 | 2.52  |
> | α=0.5 | 63.74 | 97.40 | 64.10 | 58.33 | 2.26  |
> | α=0.7  | 64.37 | 98.29 | 64.56 | 57.27 | 2.22  |
> | α=0.9   | 65.56 | 98.57 | 64.23 | 57.01 | 2.32  |
> | α=1     | 66.41 | 98.40 | 64.59 | 56.36 | 1.98  |
> | α=2     | 76.38 | 79.69 | 60.59 | 77.01 | 13.62 |
>
> **For Limitations:**
>
> **Limitations.** We will add a Limitations section: (1) Current experiments focus on CNN architectures; as demonstrated in our rebuttal, we have extended OEU to DeiT and observed consistent improvements, suggesting generalizability across architecture types. Extending to more challenging settings (larger-scale models) remains a natural direction for future work. (2) Our theoretical guarantees rely on first-order approximation, which is a standard assumption in gradient-based optimization analysis and is well-supported by our extensive empirical results.
>
> **Impact Statement.** We will acknowledge that incomplete forgetting and residual information leakage are shared risks across all approximate unlearning methods. OEU specifically mitigates these: entropy-guided unlearning drives predictions toward maximum uncertainty, inherently reducing identifiable patterns from forgotten data; gradient orthogonal projection ensures forgetting does not compromise retained knowledge, maintaining a stable and reliable unlearned model.

---

> > ### Author Rebuttal · Reviewer_AvH7 · 2026-04-03
> >
> > I appreciate the response from the authors.

---

> > > ### Author Response · Authors · 2026-04-07
> > >
> > > Thank you for your thoughtful feedback and for reviewing our rebuttal. We're pleased that our response has resolved your concerns, and we sincerely value your ongoing support.

---

### Official Review · Reviewer_6FrG · 2026-03-11

**Soundness:** 4
**Presentation:** 3
**Significance:** 3
**Originality:** 3
**Overall Recommendation:** 4
**Confidence:** 5

**Summary:**

This paper proposes Orthogonal Entropy Unlearning (OEU), a framework for machine unlearning in Quantized Neural Networks (QNNs). It replaces incorrect memorization with Entropy-Guided Unlearning (EGU) for true uncertainty and uses Gradient Orthogonal Projection (GOP) to resolve forgetting-retention conflicts. OEU consistently outperforms existing baselines, closely matching gold-standard retraining.

**Compliance With Llm Reviewing Policy:**

Affirmed.

**Final Justification:**

Based on my years of research in the field of efficient model (e.g., pruning, quantization), the novelty and workload of this submission justify its acceptance after the revisions are incorporated.

**Key Questions For Authors:**

1. While GOP is $O(|\theta|)$, computing separate gradients for $D_f$ and $D_r$ effectively doubles the backpropagation overhead per iteration. A detailed FLOPs breakdown comparing OEU to standard Fine-Tuning is necessary to quantify this increased cost.

2. Even with EGU, there is still a gap in Membership Inference Attack (MIA) performance compared to the Retrain model in some CIFAR-100 scenarios. What is the primary source of this remaining privacy leakage?

3. EGU’s high-entropy signature on the forget set may increase vulnerability to specialized Membership Inference Attacks (MIA). The authors should analyze entropy distributions across forget, retain, and test sets to ensure they are not easily distinguishable.

**Limitations:**

yes.

**Strengths And Weaknesses:**

Strengths:
1. The paper addresses a timely, practical problem at the intersection of model quantization and machine unlearning. It is well-motivated, focusing on the unique challenges of privacy compliance within the discrete parameter constraints of QNNs.

2. The combination of entropy maximization and orthogonal projection is a sophisticated response to the multi-objective nature of unlearning.

3. The identification of forgetting-misremembering conflation and directional gradient conflicts represents a significant conceptual advance. These critical, often overlooked flaws are effectively communicated through the visualizations in Figure 1.

4. GOP provides a rigorous, first-order guarantee of non-interference, which is mathematically superior to heuristics like scalar reweighting. The theoretical analysis is a strong point, lending credibility to the proposed components.



Weaknesses:

1. Theorem 4.3 relies on first-order Taylor expansion, which may be invalid for non-smooth, discrete QNNs using STE. The paper needs to justify this approximation for ultra-low precision (4/2-bit) scenarios.


2. The paper sets $\alpha=1.0$ by default, leaving its robustness and necessity unproven. A comprehensive ablation study across various $\alpha$ values (e.g., 0 to 1.0) is essential to validate performance sensitivity (FA, RA, AG) and substantiate the claimed trade-off."

3. The motivation for GOP hinges on directional conflicts (angle $> 90^\circ$). The paper should provide empirical data to confirm the frequency of these conflicts during the unlearning process in high-dimensional QNN weight spaces."


4. While OEU shows generalizability to full-precision models, the current baselines are outdated. Comparing OEU against state-of-the-art (2025-now) full-precision unlearning techniques is necessary to establish its broader significance in the field.

---

> ### Author Rebuttal · Authors · 2026-03-31
>
> **For W1:** We would like to clarify that the first-order Taylor expansion in Theorem 4.3 does not operate directly on the original discrete parameter space, but rather on the continuous latent parameter space smoothed by the STE. Specifically, although quantized weights are discrete during the forward pass, QAT maintains full-precision latent weights via the STE and updates them continuously through gradients. The STE replaces the non-differentiable quantization function with an identity estimator during backpropagation (Eq. 3), effectively defining a smooth surrogate gradient field. Under this surrogate gradient field, gradient-based analysis—including first-order Taylor expansion—follows the standard working assumption shared by all QAT methods (LSQ+, PACT, DSQ) and all gradient-based QNN unlearning methods (Q-MUL). Our analysis adopts this same assumption without introducing additional ones.
>
> Importantly, gradient orthogonality $\langle g_f^{\perp}, g_r \rangle=0$ is an exact algebraic property (Lemma A.9) independent of any approximation; the first-order term only enters when translating orthogonality into loss invariance. Empirically, small AG values under 4-bit (0.58%, CIFAR-10) and 2-bit (0.56%, SVHN) confirm reliable guidance under ultra-low precision.
>
> **For W2:** Please refer to the response to Q3 from Reviewer AvH7.
>
> **For W3:** We measure gradient conflict ratio (percentage of iterations with cos($g_f$, $g_r$)<0) on CIFAR-100 under 30% random forgetting, ResNet18. The 4-bit model shows 92.7% conflict ratio with avg cosine similarity -0.124, far exceeding full-precision (62.7%, -0.009), confirming conflicts are pervasive and amplified in QNNs, validating GOP's necessity.
>
> **For W4:**  We have added comparisons with SOTA (2025-now) techniques AMUN [1] and NatMU [2] on CIFAR-100 using ResNet18 with 4-bit. OEU consistently achieves the lowest AG.
>
> | Method  | Forgetting Rate | FA    | RA    | TA    | MIA   | AG   |
> | ------- | --------------- | ----- | ----- | ----- | ----- | ---- |
> | retrain | 10% | 74.76 | 99.98 | 72.43 | 56.36 | 0    |
> | AMUN    | 10% | 87.20 | 99.97 | 74.39 | 53.62 | 4.29 |
> | NatMU   | 10% | 78.27 | 96.92 | 74.63 | 50.35 | 3.69 |
> | OEU     | 10%  | 74.86 | 99.36 | 67.61 | 49.56 | 3.08 |
> | retrain | 30%    | 67.67 | 99.98 | 67.54 | 58.47 | 0    |
> | AMUN    | 30% | 79.89 | 97.43 | 71.75 | 54.11 | 4.61 |
> | NatMU   | 30% | 73.82 | 95.89 | 73.28 | 52.43 | 4.07 |
> | OEU     | 30%   | 66.41 | 98.40 | 64.59 | 56.36 | 1.98 |
> | retrain | 50%     | 65.40 | 99.99 | 65.74 | 68.19 | 0    |
> | AMUN    | 50%  | 76.15 | 94.31 | 70.23 | 55.89 | 8.30 |
> | NatMU   | 50%  | 71.56 | 88.21 | 68.64 | 57.44 | 7.90 |
> | OEU     | 50%   | 70.42 | 98.79 | 60.38 | 65.08 | 3.67 |
>
> [1] Ebrahimpour-Boroojeny, A., Sundaram, H., & Chandrasekaran, V. (2025, October). Not all wrong is bad: Using adversarial examples for unlearning. In Forty-second International Conference on Machine Learning.
>
> [2]  He, Z., Li, T., Cheng, X., Huang, Z., & Huang, X. (2025). Towards natural machine unlearning. IEEE Transactions on Pattern Analysis and Machine Intelligence.
>
> **For Q1:** FLOPs comparison on CIFAR-100, ResNet18, 4-bit: OEU requires 855.21 GFLOPs vs 427.54 for Fine-Tuning. The extra cost stems from separate forward/backward passes on the forget set; entropy and projection overhead is negligible (0.1236 GFLOPs). OEU converges in the same epochs (10) as other methods, with actual runtime comparable to Q-MUL and SalUn (Appendix A.3). Given OEU's superior performance, this is a reasonable trade-off.
>
> **For Q2:** The MIA gap primarily stems from the inherent difference between approximate and exact unlearning. Retrain builds a model from scratch on the retain set, so forgotten samples are statistically indistinguishable from unseen samples by construction. Approximate methods, including OEU, modify an existing model that has already learned from forgotten data, and completely eliminating all learned representations within a few fine-tuning epochs is inherently difficult. This challenge is amplified on CIFAR-100 due to its finer-grained 100-class decision boundaries, where the model encodes richer inter-class relationships during original training, making it harder to fully erase feature-level information within the constrained quantized parameter space. At lower ratios (10%), smaller forget sets provide limited gradient signal for entropy maximization.
>
> **For Q3:** We measure average output entropy across sets (ResNet18, CIFAR-100, 30%). OEU closely matches Retrain on all three, with minimal differences indicating no distinguishable high-entropy pattern.
>
> | Method  | Forget Set | Retain Set | Test Set |
> | ------- | ---------- | ---------- | -------- |
> | Retrain | 0.1842     | 0.0092     | 0.1721   |
> | SalUn   | 0.3363     | 0.2643     | 0.3156   |
> | Q-MUL   | 0.2625     | 0.1792     | 0.2455   |
> | OEU     | 0.1963     | 0.0127     | 0.1904   |

---

> > ### Author Rebuttal · Reviewer_6FrG · 2026-04-03
> >
> > I am pleased with the author's response and the additional experiments. I will maintain the current rating.

---

> > > ### Author Response · Authors · 2026-04-07
> > >
> > > We appreciate your valuable feedback and the time you dedicated to reviewing our rebuttal. We are glad to hear that your concerns have been addressed, and we truly appreciate your continued support.

---

### Official Review · Reviewer_HAFY · 2026-03-12

**Soundness:** 2
**Presentation:** 3
**Significance:** 2
**Originality:** 2
**Overall Recommendation:** 4
**Confidence:** 4

**Summary:**

Authors propose a method for machine unlearning that relies on two central ideas: 1. The entropy of model’s predictions on forget samples should be maximized. 2 Orthogonalizing the gradient updates with respect to the retain samples to prevent hurting model’s performance. Authors perform some experiments to show effectiveness of the whole pipeline while performing ablation studies to show the effectiveness of each component.

**Compliance With Llm Reviewing Policy:**

Affirmed.

**Final Justification:**

The authors have added more experiments showcasing the usefulness of their method.

**Key Questions For Authors:**

Please see weaknesses.

**Limitations:**

Yes

**Strengths And Weaknesses:**

# Strengths:

1. The paper is well-written and easy to follow.

2. The core ideas have been presented clearly and the motivation for each component are well-presented.

3. The experiments contain various model architectures and various datasets.

4. While the proposed method is general, authors additionally focus on the setting of quantized weights and activations which is less considered in works on unlearning.

# Weaknesses:

1. The target of EGU is to assign equal probability to each class. This method should work similar to the method that chooses a label for each sample in D_f uniformly at random in each epoch, given enough number of epochs (a.k.a. Random labeling)? Authors could verify that through ablation studies. In that case the contribution comes mostly from the GOP component.

2. Recent work in unlearning (which is missing from the analysis of this paper), empirically shows that the to evade the membership inference attacks, which are designed to detect training samples from unseen samples, unlearning methods should behave similar to unseen samples (test samples) on the forget samples. The behavior of a trained model on unseen samples is not predicted uniformly at random, therefore enforcing the model to behave like that will lead to detection by MIA methods because of over-unlearning phenomena which has been mentioned by recent work in unlearning [2].

3. The only privacy method used for evaluating the quality of unlearning is an outdated MIA. More recent MIAs have been shown to greatly outperform these earlier methods [3,4]. Recent unlearning methods have adapted these to the setting of unlearning evaluation [1]. There are more recent works on stronger MIAs for unlearning evaluation [5]. Since the approximate methods, such as the one presented in this paper, do not come with theoretical guarantees for unlearning, they need to utilize SOTA evaluation methods for empirical evaluations. Otherwise, they give a “false sense of privacy” [5].

4. Theoretical contributions are not significant and do not provide unlearning guarantees. The theoretical results are mostly basic known properties (e.g., maximizing entropy leads to a uniform distribution, and orthogonal projection leads to zero inner product with the projected vector) which are well-known.

5. There are recent methods in unlearning that have been shown to outperform the baselines used in this paper and yet they are missing in the experiments [1,6].

[1] Ebrahimpour-Boroojeny, A., Sundaram, H., & Chandrasekaran, V. (2025, October). Not all wrong is bad: Using adversarial examples for unlearning. In Forty-second International Conference on Machine Learning.

[2] Shi, W., Lee, J., Huang, Y., Malladi, S., Zhao, J., Holtzman, A., ... & Zhang, C. (2024). Muse: Machine unlearning six-way evaluation for language models. arXiv preprint arXiv:2407.06460.


[3] Zarifzadeh, S., Liu, P., and Shokri, R. Low-cost high-power membership inference attacks. In Forty-first International Conference on Machine Learning, 2024.

[4] Carlini, N., Chien, S., Nasr, M., Song, S., Terzis, A., & Tramer, F. (2022, May). Membership inference attacks from first principles. In 2022 IEEE symposium on security and privacy (SP) (pp. 1897-1914). IEEE.

[5] Hayes, J., Shumailov, I., Triantafillou, E., Khalifa, A., & Papernot, N. (2025, April). Inexact unlearning needs more careful evaluations to avoid a false sense of privacy. In 2025 IEEE Conference on Secure and Trustworthy Machine Learning (SaTML) (pp. 497-519). IEEE.

[6] He, Z., Li, T., Cheng, X., Huang, Z., & Huang, X. (2025). Towards natural machine unlearning. IEEE Transactions on Pattern Analysis and Machine Intelligence.

---

> ### Author Rebuttal · Authors · 2026-03-31
>
> **For W1:** We would like to clarify that  EGU and Random Labeling (RL) are fundamentally different, even given sufficient epochs.
>
> **Theoretically**, Propositions A.5-A.6 show that even with uniformly random labels each epoch, the expected target is $\bar{p}(k|x)$=0 for k=y and 1/(K−1) for k≠y— suppressing the true class to zero rather than uniform 1/K, yielding KL divergence log(K/(K−1))>0. EGU directly optimizes toward uniform 1/K for all classes including the true class, achieving $D_{KL}(\bar{p} \| \mathcal{U})$=0 (Theorem 4.2). This gap cannot be closed by more epochs.
>
> **Empirically**, "w/o EGU" in Table 2 equals RL+GOP. Additional ablations (ResNet18, CIFAR-10, 30% forget) confirm EGU+GOP consistently outperforms RL+GOP:
>
> | Method| FA| RA | TA | MIA | AG |
> | ------------- | ----- | ----- | ----- | ----- | ---- |
> | Retrain | 91.61 | 99.97 | 91.29 | 16.11 | 0 |
> | RL+GOP | 93.04 | 98.32 | 92.03 | 11.10 | 2.20 |
> | EGU+GOP (OEU) | 95.25 | 99.83 | 91.64 | 15.02 | 1.31 |
>
> This advantage holds even when $L_{retain}$ prevents the model from reaching maximum entropy — the gain stems from EGU's unbiased optimization direction toward uncertainty, in contrast to RL's  suppression of the true class toward zero.
>
> **For W2:**  We would like to clarify that our method does not lead to over-forgetting that can be detected by MIA.
>
> EGU is only one of the constraint objectives in the unlearning process. If EGU were used alone to optimize $L_{forget}$, the model's outputs on forget data would indeed become uniformly distributed, as shown in Appendix A.2. However, in the complete unlearning process, $L_{forget}$ is jointly optimized with $L_{retain}$ (Eq.11), which together constrain the model's convergence. When $L_{forget}$ pushes predictions toward higher entropy, $L_{retain}$ simultaneously pulls them back toward confident predictions (low entropy) on retain samples. Since forget and retain samples typically share similar features, the gradients from $L_{retain}$ implicitly constrain the entropy increase on forget samples. Unlike label manipulation methods that train the model to confidently predict specific incorrect classes, EGU provides a unbiased direction for unlearning in the information-theoretic sense—it guides the model away from confident predictions on forget samples without introducing bias toward any particular class.
>
> Empirical verification (CIFAR-100, ResNet-18, 30%):
>
> (a) Average Output Entropy. Please refer to the response to Q3 from Reviewer 6FrG.The retrained model itself exhibits elevated entropy on forget samples — this is natural behavior for data the model has not seen during training. OEU closely matches this entropy level rather than reaching the theoretical maximum, demonstrating that our method does not over-forget.
>
> (b)  KL divergence from Retrain: OEU achieves lowest KLD on forget (0.3961), retain (0.1926), and test sets (0.2693) vs. SalUn (0.4536/0.3379/0.3695) and Q-MUL (0.4188/0.2394/0.2967), demonstrating that OEU's outputs are most similar to those of Retrain.
>
> MIA results refute over-unlearning: if it occurred, forget samples would be anomalously distinguishable, causing large MIA gaps. Table 1 shows OEU achieves MIA gaps closest to Retrain in most cases.  (e.g., 1.64% on CIFAR-10 at 10%, 2.11% on CIFAR-100 at 30%).
>
> **For W3:** Our current MIA follows protocols of all baselines for fair comparison. Following the reviewer's suggestion, we conducted experiments with stronger MIA from [5] (ResNet18, CIFAR-100, 30%):
>
> | Method  | FA    | RA    | TA    | MIA   | AG   |
> | ------- | ----- | ----- | ----- | ----- | ---- |
> | Retrain | 67.67 | 99.98 | 67.54 | 57.61 | 0    |
> | SalUn   | 79.60 | 96.70 | 63.41 | 54.87 | 5.52 |
> | Q-MUL   | 73.90 | 97.63 | 65.80 | 62.32 | 3.76 |
> | OEU     | 66.41 | 98.40 | 64.59 | 55.12 | 2.07 |
>
> Under stronger MIA, OEU still achieves lowest AG (2.07%) and closest MIA to Retrain (gap 2.49%), confirming privacy improvements are robust rather than reflecting a false sense of privacy.
>
> **For W4:** Our contribution is not in proposing new mathematical tools, but in identifying fundamental limitations of existing unlearning paradigms for QNNs and providing a principled framework to address them. (1) We formally prove why "learning wrong answers" fails. Recognizing that entropy maximization + orthogonal projection address QNN unlearning's two core challenges is a non-trivial insight overlooked by prior work. (2) Prior works universally adopt label manipulation; our entropy-guided approach is a conceptually new direction. Q-MUL's scalar reweighting cannot resolve conflicts when gradients exceed 90°; our layer-wise normalized projection (Eq.9-10) addresses heterogeneous STE errors, with no precedent in unlearning.
>
> **For W5:** We have added recent methods (Please refer to Reviewer  6FrG W4) and will discuss [1]-[6] in detail.

---

> > ### Author Rebuttal · Reviewer_HAFY · 2026-04-04
> >
> > I want to the authors for their responses and additional experiments and evaluations. I raise my score accordingly.

---

> > > ### Author Response · Authors · 2026-04-07
> > >
> > > We're thankful for your constructive feedback and for engaging with our rebuttal. We are delighted that our rebuttal have fully addressed your concerns.

---

### Decision · Program_Chairs · 2026-04-30

**Decision:**

Accept (regular)

**Comment:**

This work addresses the problem of unlearning in deep models, that is a topic of actuality. The authors propose a method based on entropy maximization in the forget set and orthogonal projection for the gradient, to avoid interferences. The paper is clear in its intent, and the setup, although simplistic, are standard and showcase the potential of the proposed approach.

The reviews are in general positive and after rebuttal the work receives unanimous acceptance from the reviewers, as the issues raised have been promptly discussed and corrected. The authors are encouraged to include them in the revised version of their manuscript.